# The Probability Simplex is Compatible

## Abstract

Training modern AI models has become increasingly expensive, and updating these base models can significantly alter the behavior of applications or services built on them, due to changes to internal feature representations. In retrieval systems, this involves re-extracting feature vectors for all gallery data. This process can be computationally expensive and time-consuming, especially for large-scale gallery sets. To address this issue, backward compatible learning was introduced, allowing direct comparison between the representations of the base model and the updated ones. Existing backward compatible methods introduce additional losses or specific network architecture changes, which require the availability of base models, thereby limiting compatibility with models trained independently. In this paper, we show that any independently trained model can be made compatible with any other by simply using features derived from softmax outputs. We leverage the geometric properties of the softmax function, which projects vectors into the Probability Simplex, preserving the alignment of softmax outputs across model updates and verifying the definition of compatibility. A similar property is observed when using a feature representation derived from logits. They distribute in a simplex configuration, but with a wider spread in the feature distribution than softmax outputs, leading to a more robust and transferable representation. Our framework achieves state-of-the-art performance on standard benchmarks, where either the number of training classes extends across multiple steps or the base model is updated with advanced network architectures, showing that any publicly available pretrained model are compatible without requiring any additional training or adaptation. Our code will be made available upon acceptance.

## 1 Introduction

Training modern AI models has become increasingly expensive, limiting accessibility to a few well-resourced organizations (Wolf et al., 2019; Radford et al., 2021; Dubey et al., 2024; Jiang et al., 2023; Anthropic, 2024). Despite reductions in model parameters and computing costs (Dubey et al., 2024), training from scratch, fine-tuning and inference (i.e., test time) (Wang et al., 2024; OpenAI, 2024) remain economically challenging, particularly for smaller teams. As a result, these models are increasingly offered as services through APIs, a trend that is likely to continue due to persistent high costs and the significant benefits of scaling laws (Kaplan et al., 2020). Offering models as services not only simplifies the development of new applications but also enables their widespread use across various fields. However, updating these base models can completely transform the behavior of the applications or services built upon them (Raffel, 2023).

Several factors may drive updates to the base model (Raffel, 2023; Yadav et al., 2024), including evolving training strategies (Biondi et al., 2024; Echterhoff et al., 2024; Shen et al., 2020), advancements in architectures (Touvron et al., 2023), the availability of higher-quality datasets (Gunasekar et al., 2023), the expansion of training classes, or extended training periods (Biderman et al., 2023; Raffel, 2023). These advancements encapsulate rapid progress in a unified model, simplifying usage as models, datasets, and computational infrastructures grow in size and complexity (Bommasani et al., 2021; Sorscher et al., 2022).

A common scenario involves developers focusing on performance improvements in model updates, potentially compromising compatibility with earlier model versions. Concurrently, end users, such as drivers of semi-autonomous cars, often develop a mental model of the machine learning model's capabilities (Bansal et al., 2019a). As the software updates, these human-users must continually

adjust their mental model of its functionality and capabilities, a task that is not only challenging and potentially dissatisfying but also unsafe (Bansal et al., 2019b). To minimize user adjustments, both current and new software versions run in the background to compare decisions before the final deployment (Templeton, 2019). Similarly, image retrieval services might experience unexpected changes in image rankings following a model update, requiring users to adapt when previously top-ranked retrieved images no longer appear first (Shen et al., 2020). In the same vein, recent observations have shown similar issues with large language models (LLMs) (Echterhoff et al., 2024), which directly raises concerns about AI safety and alignment. This is particularly critical when decisions and textual outputs from these agents are translated into actions (Amodei et al., 2016; Ngo et al., 2024).

This general issue of updating a base model seamlessly while ensuring compatibility has been independently explored with research emphasis varying based on the type of compatibility required with earlier models: 1) if through the ultimate layer, it targets the problem of "negative flips"— ensure that the new model only mimics the old model when it is correct. (Milani Fard et al., 2016; Yan et al., 2021); 2) if through the penultimate layer, it involves learning "backward-compatible" feature representations that can be interchangeably used across models (Shen et al., 2020; Biondi et al., 2023); and 3) if through the whole architecture it targets merging by averaging earlier models using the concept of "linear mode connectivity" (Ainsworth et al., 2023; Wortsman et al., 2022; Matena & Raffel, 2022; Frankle et al., 2020).

Although repositories of pre-trained foundation models are publicly available (Wolf et al., 2020), merging large models still requires not only the same architecture but also access to weights behind an API and substantial compute resources (Yadav et al., 2024). Backward compatibility, as defined in (Shen et al., 2020) originally developed to avoid image re-indexing in retrieval, is crucial in overcoming these challenges by ensuring models remain effective on targeted tasks and compatible with established input and output (Raffel, 2023). However, most existing studies on backward compatibility require the availability of previous or base models, thereby limiting compatibility with models trained independently (Shen et al., 2020; Meng et al., 2021; Duggal et al., 2021). Recent studies have begun to address this issue (Biondi et al., 2023; 2024). Although some theoretical results support these methods, implementation requires agreement on using a specific fixed classifier among parties before training the models (Pernici et al., 2021; Zhu et al., 2021). This agreement is often challenging due to competition-related issue between organizations, and the fixed classifier requires substantial pre-allocation for future classes.

In this paper, we demonstrate that any independently trained model can be made compatible with any other, and can remain compatible when expanded with new classes. This is achieved by using feature representations derived from softmax outputs. The softmax function projects the feature space into the Probability Simplex, a geometric configuration in which the vertices—corresponding to the standard unit vector of the feature space—are maximally equidistant. Although the Probability Simplex evolves with the introduction of new classes, we show that a projection matrix can be defined to preserve alignment across model updates, resulting in stationary representations that verify the definition of compatibility in Shen et al. (2020). A similar property is observed when using feature representations derived from logits. We demonstrate that, during training, they distribute in the same simplex configuration of the softmax outputs. However, they present a wider spread in the feature distribution than softmax outputs, leading to a more robust and transferable representation.

Our framework achieves state-of-the-art performance on standard benchmarks, where either the number of training classes grows over multiple steps or the base model is upgraded with more advanced architectures. This shows that any publicly available pretrained model can be seamlessly made compatible without requiring additional training or adaptation.

Our main contributions are threefold:

1. We introduce a novel approach to model compatibility that requires no training and utilizes independently trained models.

2. Theoretical support for the methodology is provided, confirming its validity and broad applicability.

3. We demonstrate significant empirical improvements, particularly in scenarios with frequent model updates.

## 2 RELATED WORKS

The aim of compatible training is to learn representations that can be used interchangeably when updating a model, thereby avoiding the re-indexing of the gallery set in retrieval. This basic formulation, originally proposed in Shen et al. (2020), is in principle a practical method for evaluating model updates, even when the base or updated models are not explicitly used for retrieval but operate as input-output black boxes. Backward Compatible Training (BCT) was introduced by Shen et al. (2020), employing the previous classifier as a fixed reference during the training of the new model, so that new feature vector can align to their old class prototypes. Following works introduced additional regularization loss functions (Meng et al., 2021; Zhang et al., 2021; 2022; Pan et al., 2023) to align the updated representation with the old one. In particular, in Meng et al. (2021) (LCE) a combination of loss functions is used to align class means between different model upgrades and to achieve more compact intra-class distributions while learning new data, directly optimizing one of the inequalities in the definition of compatibility given by Shen et al. (2020). An adversarial learning discriminator was introduced in BCT by Pan et al. (2023) (AdvBCT) to minimize the distribution disparity between features from the old and new models.

Due to the regularization constraints imposed to achieve compatibility, the performance of the new backward-compatible model often does not reach that of a newly independently trained model (Zhou et al., 2023). To address this issue, other methods achieved compatibility between independently trained models using mapping-based approaches (Chen et al., 2019; Wang et al., 2020; Meng et al., 2021; Ramanujan et al., 2022; Jaeckle et al., 2023). However, learning these mapping functions on top of the models introduces additional computational overhead, which can become infeasible with large datasets, and involves the composition of these modules when the model is updated several times. In a similar vein, in Zhou et al. (2023); Ricci et al. (2024), at each model update, the feature space of the model is expanded to obtain compatibility with the previous model while learning the new information in the expanded part of the feature space. Although these methods can leverage the discriminative power of a newly independently trained model, their implementations require specific network architecture changes.

The solution proposed by Biondi et al. (2023) (CoReS) to achieve compatibility between independently trained models involves learning according to a pre-allocated $d$-Simplex fixed classifier. This approach learns stationary representations, with features remaining aligned with their fixed class prototypes, while new classes are incorporated into pre-allocated regions of the feature space. Our formulation leverages one-hot encoded label vectors as a fixed reference, implicitly supporting stationarity in the alignment of features derived from softmax outputs. The implicit use of one-hot encoded label vectors as a fixed reference obviates the need for a unified fixed classifier among parties before training the models, given that the softmax function is inherently universal across these parties.

## 3 THE PROBABILITY SIMPLEX LEADS TO COMPATIBLE REPRESENTATIONS

### 3.1 PRELIMINARIES ON BACKWARD-COMPATIBLE REPRESENTATION LEARNING

Let $\mathcal{G} = \{(\mathbf{x}_i, y_i)\}_{i=1}^{N_g}$ be the gallery set composed of $N_g$ images $\mathbf{x}_i$, each belonging to class $y_i$ and a query set $\mathcal{Q} = \{\mathbf{x}_i\}_{i=1}^{N_q}$ composed of $N_q$ images $\mathbf{x}_i$. A base model indexes the gallery set by extracting feature vectors for each image, which are then used to perform retrieval tasks with the feature vectors from the query set.

At a given time step $t$, the *base model* can be updated to include new network architectures or to increase the number of training classes. At this time step $t$, the training set, denoted $\mathcal{D}^t = \{(\mathbf{x}_i, y_i)\}_{i=1}^{N^t}$, comprises of $N^t$ labeled images $\mathbf{x}_i$, where each label $y_i$ corresponds to one of the $C^t$ classes. Specifically, when the number of classes increases, the base model is updated using the dataset $\mathcal{D}^t = \mathcal{D}^{t-1} \cup \mathcal{X}^t$, where $\mathcal{D}^{t-1}$ represents the existing data up to step $t-1$, and $\mathcal{X}^t$ includes the new data for step $t$.

Backward compatibility between the *updated model* at time step $t$ and the base model learned at a previous step $k$ is achieved if the features of the queries extracted by the current model, $\Phi_{\mathcal{Q}}^t = \{\mathbf{h}_i^t \in \mathbb{R}^d \mid \forall \mathbf{x}_i \in \mathcal{Q}\}$, can be compared with the gallery features obtained by the old model $\Phi_{\mathcal{G}}^k = \{\mathbf{h}_i^k \in \mathbb{R}^d \mid \forall \mathbf{x}_i \in \mathcal{G}\}$, while preserving the accuracy and avoiding the necessity of re-extracting

the gallery features using the updated model. Here, $\mathbf{h}_i^t \in \mathbb{R}^d$ represents the feature vector of the image $\mathbf{x}_i$ extracted with the updated model at step $t$ where $d$ is the dimension of the feature space.

Following the outlined setup, the formal definition of backward-compatible representations by Shen et al. (2020) specifies that:

**Definition 1** (*Compatibility*). *The representation of a base model learned at step $k$ is compatible with the representation of the updated model learned at step $t$, with $k < t$, if it holds that:*

$$d\big(\mathbf{h}_i^k, \mathbf{h}_j^t\big) \leq d\big(\mathbf{h}_i^k, \mathbf{h}_j^k\big), \qquad \forall (i,j) \in \{(i,j)|y_i = y_j\} \tag{1a}$$

$$d\big(\mathbf{h}_i^k, \mathbf{h}_j^t\big) \geq d\big(\mathbf{h}_i^t, \mathbf{h}_j^t\big), \qquad \forall (i,j) \in \{(i,j)|y_i \neq y_j\} \tag{1b}$$

*where $d(\cdot, \cdot)$ is a distance function and $y_i$ and $y_j$ are class labels associated to $\mathbf{h}_i$ and $\mathbf{h}_j$, respectively.*

The inequalities of Def. 1 specifies that the updated model representation performs at least as well as the base model in separating images from different classes and grouping those from the same classes.

## 3.2 PROBABILITY SIMPLEX PROJECTIONS (PSP)

In this section, we demonstrate that the softmax output from independently trained models provides compatible representations according to Def. 1. To this aim, we consider the geometric properties of the softmax function. This function maps the vector space $\mathbb{R}^C$ onto the standard $(C-1)$-simplex, resulting in a dimensional reduction of one, due to the linear constraint that requires all output values to sum to 1. Consequently, this confines the output to reside in a $(C-1)$-dimensional hyperplane within the $C$-dimensional space. We demonstrate that this mapping inherently provides the basis for defining a rigorous methodology where adding new classes to a base model not only preserves the alignment of softmax vectors after a model update, but also meets the second criterion of compatibility in Def. 1, recently shown to be not strictly feasible by Biondi et al. (2024).

Given a base model at the time step $k$, we consider the feature representation $\mathbf{h}^k$ as the normalized softmax output vector subjected to the projection $\mathbf{P}_{k,k}$, given by the equation:

$$\mathbf{h}^k = \frac{\mathbf{P}_{k,k}\, \sigma(\mathbf{z}^k)}{\|\mathbf{P}_{k,k}\, \sigma(\mathbf{z}^k)\|_2}, \quad \mathbf{h}^k \in \mathbb{R}^{C^k}, \tag{2}$$

where $\sigma(\cdot)$ represents the softmax function, $\mathbf{z}^k = \mathbf{W}^k \cdot \phi^k(\mathbf{x})$ denotes the logit output, $\phi^k$ the base model and $\mathbf{W}^k$ its classifier matrix. The projection $\mathbf{P}_{k,k}$ transforms the softmax output so that it is centered at the origin of the axes, allowing its mapping onto the hypersphere through normalization.

Thus, Eq. 2 transforms softmax probability vectors into hyperspherical feature representation vectors. Given a reasonably low training error of a model, the probability outputs of the softmax function are expected to cluster near the vertices of the Probability Simplex. After projection with $\mathbf{P}_{k,k}$, the normalization operation further projects the softmax probability outputs onto the hypersphere, forming a distribution that closely approximates a von Mises-Fisher distribution ——the hyperspherical analog of the Normal distribution. This approximation enables us to derive a closed-form solution, which is utilized in Theorem 1, to analytically determine the expected distance required in Def. 1 to evaluate and verify compatibility. Specifically, the projection $\mathbf{P}_{k,k}$, derived from the centering matrix as described in Marden (1996), is defined by:

$$\mathbf{P}_{k,k} = \mathbf{I}_{C^k} - \frac{1}{C^k}\mathbf{J}_{C^k} \tag{3}$$

where $\mathbf{I}_{C^k}$ is the identity matrix for $C^k$ classes, and $\mathbf{J}_{C^k}$ is a $C^k \times C^k$ matrix entirely composed of ones. The projection $\mathbf{P}_{t,k}$ can be then defined as:

$$\mathbf{P}_{t,k} = [\mathbf{P}_{k,k} \mid \mathbf{0}] \tag{4}$$

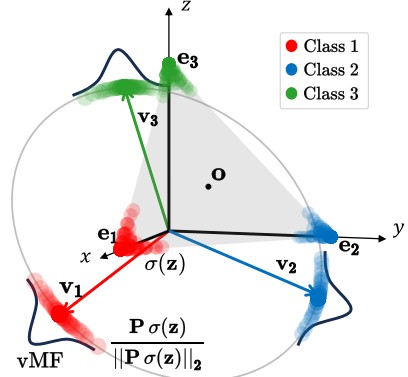

Figure 1: Softmax outputs, represented as colored points in the Probability Simplex, are projected onto the hypersphere (shown as colored points on the hypersphere). Each class's softmax features, approximated by the von Mises-Fisher distribution, are centered on the class prototypes as defined in Eq. 5 (illustrated with colored vectors).

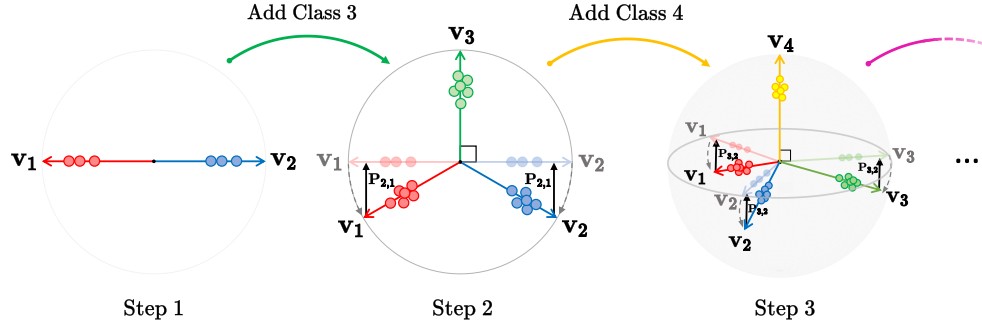

Figure 2: Visualization of Probability Simplex Projections (PSP) across two model updates. From left to right: a 1-simplex, a 2-simplex with a new orthogonal class prototype (green), and a 3-simplex with a new orthogonal class prototype (yellow). Gray dashed lines show how class prototypes (colored lines) are moved accordingly to form the new simplex structure, while black solid lines demonstrate their alignment with the old class prototypes up to a projection $\mathbf{P}_{t,k}$. This alignment leads to compatible representations.

where $\mathbf{0} \in \mathbb{R}^{(C^t - C^k) \times C^k}$ is the zero matrix with $C^t$ and $C^k$ the number of classes of the updated and base model, respectively. Prop. 2 demonstrates that $\mathbf{P}_{t,k}$ provides a simple yet surprisingly beneficial capability, enabling the projection of softmax outputs from the representation space of the updated model to that of the base model. This unifies the representation spaces prior to normalization onto the hypersphere providing an analytical treatment of the expanding feature space, showing its inherent compatibility as defined in Def. 1. Given these considerations we refer to our methodology as Probability Simplex Projections (PSP). Before discussing the theorems central to our framework, we review the definition of the Probability Simplex, which has been adapted to accommodate a varying number of classes, denoted as $C^k$, present in the model at time $k$:

**Definition 2** (**Probability Simplex**). *Let $\Delta^{C^k-1}$ be the $(C^k - 1)$-dimensional simplex in $\mathbb{R}^{C^k}$. Its vertices are the standard unit vectors $\mathbf{e}_1, \mathbf{e}_2, \ldots, \mathbf{e}_{C^k}$, and its center is $\mathbf{o}^k = \frac{1}{C^k} \sum_{i=1}^{C^k} \mathbf{e}_i$. The Probability Simplex is defined as: $\Delta^{C^k-1} = \{\mathbf{u} \in \mathbb{R}^{C^k} \mid \sum_{i=1}^{C^k} u_i = 1, \, u_i \geq 0 \, \forall i, 1 \leq i \leq C^k\}$.*

In essence we aim to induce hyperspherical stationarity in the softmax layer. This is established by considering the abstract simplex prototypes in the projection $\mathbf{P}_{k,k}$ of Eq. 3 as:

$$\mathbf{v}_y^k = \mathbf{e}_y - \mathbf{o}^k \quad \forall y, 1 \leq y \leq C^k. \tag{5}$$

According to Eq. 5, the one-hot encoded label vectors serve as a fixed reference for the class prototypes. Despite $\mathbf{v}_y^k$ subjected to the shift of $\mathbf{o}^k$ with each update, we demonstrate that its projection retains a fixed position, ensuring stationary references for representation in Eq. 2. The projected vector $\mathbf{v}_y^k$ operates as a fixed unified reference to which the normalized projected softmax outputs align. Furthermore, the projection $\mathbf{P}_{t,k}$ of Eq. 4 transforms the softmax outputs of step $t$ and locate them around $\mathbf{v}_y^k$ of Eq. 5, which can be interpreted as the mean direction of the von Mises-Fisher distribution on the hypersphere. We present our framework through the following propositions and a subsequent final theorem. The first proposition demonstrates the orthogonality of new classes with respect to old ones by guaranteeing zero projection of the newly added class prototypes relative to the previous ones. In the second proposition, we demonstrate that alignment across expanded models can be achieved through a projection matrix. The final theorem shows how compatibility can be achieved—verifying the compatibility requirements (including the second constraint)— using this representation as they distribute in a unified simplex configuration with fixed references provided by Eq. 5. Fig. 2 illustrates the basic geometry forming the basis of our theoretical approach. The figure shows how the prototype vectors interact with the normalized softmax outputs, providing a visualization of the fixed references to which the projected normalized softmax features align.

The updating process of a base model results in an extension of the classifier to accommodate additional outputs for new classes. This involves an orthogonal expansion of the one-hot encoded labels corresponding to the vertices of the newly added classes, causing a change in the configuration of the updated class prototypes of the old classes (as illustrated by the dashed lines in Fig. 2). This concept is formalized in the following proposition:

**Proposition 1.** *Assuming an increase in the number of classes to $C^t$ from $C^k$ at a new step $t$, the class prototypes of the newly added classes are orthogonal with respect to the prototypes from the previous step $k$.*

*Proof.* The proof of this proposition is provided in Appendix A. □

As demonstrated in Prop. 1, the simplex's symmetry ensures orthogonality of the representation of the new added classes with the old ones. This implies that the new information introduced by these classes is also orthogonal to the previous information, ensuring no interaction between them. This allows us to formulate the following proposition:

**Proposition 2.** *Let $\mathbf{v}_y^t \in \mathbb{R}^{C^t}$ be the prototype vector for class $y$ within the Probability Simplex $\Delta^{C^t-1}$. Define its projection as $\mathbf{u}_y^t = \mathbf{P}_{t,k}\mathbf{v}_y^t$, where $\mathbf{P}_{t,k} = [\mathbf{V}^k|\mathbf{0}]$ with $\mathbf{V}^k = [\mathbf{v}_y^k]_{y=1}^{C^k} \in \mathbb{R}^{C^k \times C^k}$ and $\mathbf{0} \in \mathbb{R}^{(C^t-C^k) \times C^k}$ is a zero matrix. Then the resulting projected prototype $\mathbf{u}_y^t$ is aligned with the prototype $\mathbf{v}_y^k$ within $\Delta^{C^k-1}$.*

*Proof.* The proof of this proposition is provided in Appendix B. □

As consequence, class prototypes can be projected according to $\mathbf{P}_{t,k}$ from the new feature space $\mathbb{R}^{C^t}$ to the old feature space $\mathbb{R}^{C^k}$ (as illustrated by the solid lines in Fig. 2), resulting aligned with the old prototypes of the same classes. The section concludes with the compatibility theorem showing that PSP features are inherently compatible. In the derivation of the following theorem, we base our conclusions on several key assumptions: (1) The normalized projected softmax class features are assumed to follow a von Mises-Fisher (vMF) distribution; (2) the variance of this vMF distribution is assumed to decrease with each model update; and (3) our findings are established on average, verifying that same-class features are closer and different-class features are farther apart using the updated model. Further assumptions will be introduced in the proof to facilitate the progression of the mathematical analysis.

**Theorem 1** (*Probability Simplex Compatibility Theorem*). *Assuming the number of classes changes from $C^k$ at step $k$ to $C^t$ at step $t$, where $C^t \geq C^k$, the normalized softmax features of two models, independently trained at these respective steps, are compatible as formulated in Def. 1.*

*Proof.* The proof of this theorem is provided in Appendix C. □

In particular, it results that compatibility requires an angle greater than $\pi/2$ between two different classes to satisfy the second inequality in Def. 1. This requirement is always met by representations arranged in simplex geometry, as described in Def. 2.

Appendix F (Alg. 1) provides pseudocode for computing PSP feature vectors, detailing the input requirements, the construction of the projection matrix, and the normalization of the output.

### 3.3 LOGITS SIMPLEX PROJECTIONS

In this section, we show that using representations derived from logits exhibit similar properties as the normalize softmax outputs, offering an alternative representation that achieve compatibility. Thus, our framework as described in Sec. 3.2 is also valid when considering logits as features, i.e.,

$$\mathbf{h}^k = \frac{\mathbf{z}^k}{\|\mathbf{z}^k\|_2} \,, \quad \mathbf{h}^k \in \mathbb{R}^{C^k} \qquad (6)$$

This representation is named Logit Simplex Projections (LSP). Fig. 3 illustrates the configuration of softmax outputs and logits in a ResNet18 model trained on three CIFAR100 classes The figure illustrates the softmax outputs within the Probability Simplex (shown in darker gray) in $\mathbb{R}^3$, which tend to concentrate near the vertices corresponding to the canonical basis vectors of $\mathbb{R}^3$. We demonstrate that as the softmax probabilities

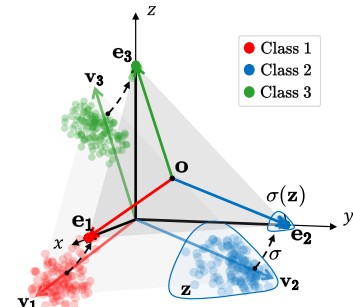

Figure 3: As softmax outputs converge towards the vertices of the Probability Simplex (in darker gray), logits configure in a simplex configuration (in lighter gray).

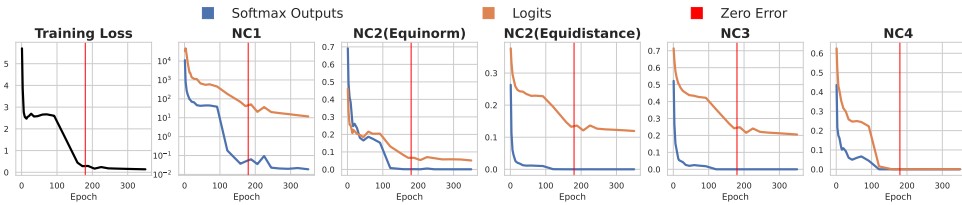

Figure 4: Neural Collapse hypothesis evaluated using feature representation derived from softmax outputs (blue lines) and logits (orange lines) on CIFAR100 with ResNet18.

converge towards these vertices, logits configure into a simplex defined by the "abstract" class prototypes of Eq. 5. The following proposition formally establishes this result.

**Proposition 3.** *As the softmax outputs approach the vertices of the Probability Simplex, the corresponding logits vectors assume a simplex configuration, with class prototypes aligning to the vectors specified in Eq. 5.*

*Proof.* The proof of this proposition is provided in Appendix D. □

From Prop. 3, it directly follows that logits exhibit the same geometrical properties as normalized softmax outputs, as specified in Eq. 2, and thus benefit from the same results described in Theorem 1. To empirically verify the theoretical results of Prop. 3, we utilize the Neural Collapse hypothesis (NC) introduced by Papyan et al. (2020). This hypothesis provides a methodology for examining how PSP/LSP features collapse to the abstract prototypes we have defined in Eq. 5. Specifically, NC hypothesis evaluates: the within-class covariance of features (NC1); the formation of a simplex where vertices have equal norm (NC2 equinorm hypothesis) and are at the maximum possible distance from each other (NC2 equidistance hypothesis); the convergence of class means to class prototypes (NC3); and the applicability of the nearest class-means rule for classifying features (NC4).

Fig. 4 displays training loss and the NC hypothesis values for softmax outputs (blue curves) and logits (orange curves) of a ResNet18 network trained on CIFAR100. The NC2 equidistance and NC3 curves indicate that logits are less collapsed compared to the softmax outputs, suggesting reduced alignment. Conversely, the NC1 values show that logits present a higher spread of feature distribution than softmax outputs, as this metric assesses the extent of within-class covariance of features.

The higher spread of logits highlights an inherent trade-off between alignment and spread in the feature distribution similar to that described in Chen et al. (2022), indicating that achieving both simultaneously is challenging. Although alignment is beneficial for achieving compatible representations as demonstrated in Theorem 1, a wider spread is desirable to obtain a representation that is transferable and robust (Wang & Isola, 2020). Using features derived from logits as in Eq. 6 provides a better balance between alignment and spread compared to the representation of Eq. 2. In Appendix E, we show that this effect occurs in several neural network architectures (ResNet18, ResNet50, and DenseNet) and datasets (CIFAR100 and TinyImageNet200).

As for PSP, Appendix F (Alg. 1) provides pseudocode for computing LSP feature vectors.

## 4 EXPERIMENTAL RESULTS

We conducted extensive experiments to demonstrate the performance of our framework, which uses features derived from softmax outputs (PSP) or logits (LSP).

### 4.1 COMPATIBILITY METRICS

The definition of compatibility in Def. 1 requires evaluating every pair-wise distance between datapoints in the dataset, which becomes computationally challenging as the dataset size increases. According to this, the updated model learned at step $t$ is said to be backward-compatible with the base model learned at step $k$ if the Empirical Compatibility Criterion Shen et al. (2020) holds

$$M\left(\Phi_t^{\mathcal{Q}}, \Phi_k^{\mathcal{G}}\right) > M\left(\Phi_k^{\mathcal{Q}}, \Phi_k^{\mathcal{G}}\right) \quad \text{with } t > k, \tag{7}$$

where $M$ is a performance metric, $M\left(\Phi_t^{\mathcal{Q}}, \Phi_k^{\mathcal{G}}\right)$ denotes the cross-test where gallery features are obtained with the updated model of step $t$ and query features with the base model of step $k$, and

$M\left(\Phi_k^{\mathcal{Q}}, \Phi_k^{\mathcal{G}}\right)$ the self-test where both gallery and query features are obtained with model of step $k$. Given $T$ models, the Compatibility Matrix $\mathbf{C} \in \mathbb{R}^{T \times T}$ (Biondi et al., 2023) represents all the models combinations across multiple $T$ steps. $\mathbf{C}$ is defined as

$$\mathbf{C}_{t,k} = \begin{cases} 0 & \text{if } t < k \\ M\left(\Phi_k^{\mathcal{Q}}, \Phi_k^{\mathcal{G}}\right) & \text{if } t = k \\ M\left(\Phi_t^{\mathcal{Q}}, \Phi_k^{\mathcal{G}}\right) & \text{if } t > k \end{cases} \tag{8}$$

Self-tests are reported in the main diagonal, while cross-tests in the lower sub-diagonal values. To effectively evaluate multiple updates of large compatibility matrices, we adopted the metrics from the recent paper (Biondi et al., 2023). These metrics, reported below, "summarize" the performance in the compatibility matrix by extracting a single number that represents overall performance, aiming for a balance between being overly compatible and maintaining expressiveness, allowing the model to fully leverage newly learned information without being constrained by the outdated, lower-performing model. Based on these observations, the following scalar metrics are used to evaluate compatibility and accuracy across $T$ steps of compatible learning:

- *Average Compatibility*: $AC = \frac{2}{T(T-1)} \sum_{t=2}^{T} \sum_{k=1}^{t-1} \mathbb{1}\left(\mathbf{C}_{t,k} > \mathbf{C}_{k,k}\right)$, being $\mathbb{1}$ is the indicator function. It represents the normalized count of times compatibility is achieved over $T$ steps according to Eq. 7. $AC$ shows how often compatibility is achieved across $T$ model updates.

- *Average Accuracy*: $AA = \frac{2}{T(T+1)} \sum_{t=1}^{T} \sum_{k=1}^{t} \mathbf{C}_{t,k}$ expressing the average accuracy (in term of $M$) over $T$ steps, considering all the self-tests and cross-tests.

- *Average Compatibility Accuracy*: $ACA = \frac{2}{T(T-1)} \sum_{t=2}^{T} \sum_{k=1}^{t-1} \mathbf{C}_{t,k}$ s.t. $\mathbf{C}_{t,k} > \mathbf{C}_{k,k}$ expressing the average accuracy (in term of $M$) of cross-tests that satisfy Eq. 7 over $T$ steps. This metric computes the average accuracy only when compatibility is achieved.

In our experiments, we use the Recall@1 as performance metric $M$ according to the cosine similarity between query features $\Phi_{(\cdot)}^{\mathcal{Q}}$ and gallery features $\Phi_{(\cdot)}^{\mathcal{G}}$.

## 4.2 COMPARATIVE RESULTS

We performed a comparative analysis of PSP and LSP under two distinct scenarios. In the first scenario, we evaluated compatibility across standard benchmarks, where the number of training classes is extended across multiple steps. In the second scenario, we assessed compatibility of publicly available pretrained models with an advanced network expressiveness across multiple steps.

**Extended Class Results.** Tab. 1 presents the compatibility performance for learning scenarios in which the base model is updated with an extended number of training classes for each new step. In these experiments, we compared PSP and LSP with BCT (Shen et al., 2020), CoReS (Biondi et al., 2023), LCE (Meng et al., 2021), AdvBCT (Pan et al., 2023), and with a baseline method where features are derived from the encoder output of the base model. We have also compared with two approaches where the classifier of the base model follows a fixed Equiangular Tight Frame (ETF) (Papyan et al., 2020) configuration where classes are pre-allocated (Yang et al., 2022) and the base models is trained under cross-entropy loss (ETF-CE) or the dot-regression loss (ETF-DR). All experiments are conducted using the public implementations of the methods on a Nvidia Quadro RTX A6000 with 24GB and two Nvidia A100 GPUs, each with 40GB.

In Tab. 1a, we report performance on the CIFAR100 (Krizhevsky, 2009) test set, with a ResNet18 network trained for 2, 5, 20, and 50 steps using the CIFAR100 training set. Tab. 1b shows compatibility performance on the TinyImageNet200 (Le & Yang, 2015) test set, with a ResNet18 network trained for 2, 5, 20, and 50 steps using the TinyImageNet200 training set. Tab. 1c presents performance on the ImageNet1k and Google Landmark v2 (Weyand et al., 2020) test sets, where the ResNet50 and ResNet18 networks are trained for 2 and 5 steps using the ImageNet1k and Google Landmark v2 training sets, respectively. Each dataset is divided into steps, each with an equal number of training classes, i.e., $|\mathcal{X}_t| = C/T$ for $t = 1, 2, \ldots, T$, where $C$ represents the total number of classes in the dataset. More info about datasets and implementation details are in Appendix G.

Overall, Tab. 1 shows that PSP and LSP achieve state-of-the-art results, confirming our theoretical analysis. PSP and LSP outperform other approaches in terms of both $AC$ and $ACA$, particularly in

Table 1: Experimental results on CIFAR100, TinyImageNet, ImageNet, and Google Landmarks datasets in the case of extended training classes at each model update, evaluated using the $AC$, $AA$, and $ACA$ metrics, with Recall@1 as the performance metric ($M$). Dark blue numbers indicate the highest values, while light blue the second-highest values for each metric and at each step value. "nan" indicates a training error with non-numeric values; "oom" means training requires extra GPU memory; "×" denotes values cannot be computed due to method limitations.

(a) CIFAR100.

| Method | 2 steps | | | 5 steps | | | 20 steps | | | 50 steps | | |
|---|---|---|---|---|---|---|---|---|---|---|---|---|
| | $AC$ | $AA$ | $ACA$ | $AC$ | $AA$ | $ACA$ | $AC$ | $AA$ | $ACA$ | $AC$ | $AA$ | $ACA$ |
| Baseline | 0 | 29.63 | 0 | 0 | 13.60 | 0 | 0 | 04.22 | 0 | 0 | 02.29 | 0 |
| BCT (Shen et al., 2020) | 1 | 46.40 | 50.21 | 0.40 | 29.93 | 11.78 | 0.13 | 19.47 | 05.20 | 0.01 | 15.78 | 0.84 |
| CoReS (Biondi et al., 2023) | 0 | 38.75 | 0 | 0 | 29.63 | 0 | 0.07 | 24.06 | 00.26 | ¡0.01 | 22.87 | 0.27 |
| ETF-CE (Yang et al., 2022) | 0 | 38.37 | 0 | 0 | 27.26 | 0 | 0 | 20.43 | 0 | 0 | 19.43 | 0 |
| ETF-DR (Yang et al., 2022) | 0 | 36.04 | 0 | 0 | 24.66 | 0 | 0 | 18.83 | 0 | 0 | 16.87 | 0 |
| LCE (Meng et al., 2021) | 1 | 43.48 | 40.71 | 0.10 | 32.63 | 04.63 | 0 | 20.95 | 00.25 | 0 | 13.81 | 0.03 |
| AdvBCT (Pan et al., 2023) | 0 | 35.32 | 0 | 0.40 | 26.13 | 11.67 | 0.02 | 19.79 | 00.19 | 0 | 14.70 | 0.03 |
| PSP | 1 | 36.31 | 29.05 | 0.90 | 26.04 | 21.76 | 0.52 | 20.56 | 12.99 | 0.39 | 19.42 | 10.76 |
| LSP | 1 | 41.14 | 36.38 | 0.70 | 30.36 | 21.91 | 0.44 | 24.11 | 13.26 | 0.36 | 22.68 | 11.25 |

(b) TinyImageNet200.

| Method | 2 steps | | | 5 steps | | | 20 steps | | | 50 steps | | |
|---|---|---|---|---|---|---|---|---|---|---|---|---|
| | $AC$ | $AA$ | $ACA$ | $AC$ | $AA$ | $ACA$ | $AC$ | $AA$ | $ACA$ | $AC$ | $AA$ | $ACA$ |
| Baseline | 0 | 21.72 | 0 | 0 | 09.61 | 0 | 0 | 02.86 | 0 | 0 | 01.43 | 0 |
| BCT (Shen et al., 2020) | 1 | 35.29 | 37.89 | 1 | 24.42 | 22.92 | 0.64 | 18.17 | 14.75 | 0.08 | 15.41 | 02.29 |
| CoReS (Biondi et al., 2023) | 0 | 27.60 | 0 | 0.60 | 21.55 | 12.69 | 0.62 | 17.82 | 09.99 | 0.55 | 17.06 | 09.79 |
| ETF-CE (Yang et al., 2022) | 0 | 29.46 | 0 | 0.20 | 21.66 | 05.66 | 0.12 | 17.22 | 03.68 | 0.03 | 16.14 | 01.01 |
| ETF-DR (Yang et al., 2022) | 0 | 29.90 | 0 | 0 | 21.34 | 0 | 0.04 | 16.58 | 01.47 | 0.05 | 15.66 | 01.81 |
| LCE (Meng et al., 2021) | 1 | 32.11 | 30.37 | 0.60 | 24.48 | 16.92 | 0.02 | 16.51 | 00.90 | 0 | 11.10 | 0 |
| AdvBCT (Pan et al., 2023) | 0 | 24.90 | 0 | 0.70 | 18.99 | 13.93 | 0.26 | 14.65 | 04.41 | 0 | 09.34 | 0 |
| PSP | 1 | 29.88 | 25.05 | 0.90 | 21.93 | 17.99 | 0.91 | 17.53 | 15.33 | 0.90 | 16.63 | 14.95 |
| LSP | 1 | 32.44 | 29.26 | 1 | 25.10 | 23.48 | 0.83 | 20.46 | 17.34 | 0.80 | 19.48 | 16.01 |

(c) Large scale datasets: ImageNet1k and Google Landmark v2.

| Method | ImageNet1k | | | | | | Google Landmark v2 | | | | | |
|---|---|---|---|---|---|---|---|---|---|---|---|---|
| | 2 steps | | | 5 steps | | | 2 steps | | | 5 steps | | |
| | $AC$ | $AA$ | $ACA$ | $AC$ | $AA$ | $ACA$ | $AC$ | $AA$ | $ACA$ | $AC$ | $AA$ | $ACA$ |
| Baseline | 0 | 37.52 | 0 | 0 | 16.18 | 0 | 0 | 07.66 | 0 | 0 | 07.87 | 0 |
| BCT (Shen et al., 2020) | 1 | 57.53 | 58.10 | 0.20 | 34.96 | 5.28 | 1 | 10.98 | 09.12 | 0 | 08.46 | 0 |
| CoReS (Biondi et al., 2023) | 0 | 46.11 | 0 | 0 | 37.67 | 0 | oom | oom | oom | oom | oom | oom |
| ETF-CE (Yang et al., 2022) | 0 | 48.67 | 0 | 0 | 33.52 | 0 | × | × | × | × | × | × |
| ETF-DR (Yang et al., 2022) | 0 | 46.35 | 0 | 0 | 31.27 | 0 | × | × | × | × | × | × |
| LCE (Meng et al., 2021) | 0 | 47.22 | 0 | 0 | 32.67 | 0 | 1 | 09.41 | 08.03 | 0.40 | 10.31 | 3.80 |
| AdvBCT (Pan et al., 2023) | nan | nan | nan | nan | nan | nan | 1 | 11.12 | 09.67 | 0 | 10.73 | 0 |
| PSP | 1 | 49.10 | 39.80 | 1 | 35.20 | 31.09 | 1 | 08.74 | 08.24 | 0.90 | 08.53 | 7.21 |
| LSP | 1 | 50.73 | 45.62 | 0.50 | 38.03 | 14.28 | 1 | 09.31 | 08.36 | 0.80 | 09.36 | 7.34 |

the challenging scenarios with a large number of model updates. Notably, using logits as features (LSP) generally achieves higher $AA$ than using features derived from softmax outputs (PSP), but with lower $AC$ values. This is attributed to the wider spread in the feature distribution of logits, which yields higher open-set accuracy but less compatibility due to reduced alignment with the simplex configuration. Consequently, LSP often achieves higher $ACA$ values than PSP. In Appendix H, we describe how the compatibility matrix facilitates a detailed, step-by-step examination of incremental performance and trade-offs between compatibility and retrieval accuracy.

Other methods, while reporting comparable $AC$ performance with PSP and LSP with small number of model updates, exhibit a clear performance decay when the number of steps increases. This can be due to the interaction that the updated models have with the old ones, observed in both the case of regularization-based methods like BCT and mapping-based methods like LCE. In scenarios with a small number of tasks, these methods report higher $AA$, likely due to the lower model expressiveness of PSP and LSP due to their smaller representation spaces. Notably, the size of PSP and LSP representations grows linearly with the number of classes. In Appendix K, we explore the impact of dimensionality reduction on PSP and LSP representation sizes to mitigate this effect.

Table 2: Experimental results on ImageNet1k, CIFAR100 and Places365 in the case of advanced network architectures (AlexNet, ResNet50, RegNetX_3.2GF, ResNet152, and MaxViT_T), evaluated using the $AC$, $AA$, and $ACA$ metrics, with Recall@1 as the performance metric $(M)$. Models are pretrained on ImageNet1k. Dark blue numbers indicate the highest values, while light blue the second-highest values.

| Features derived from | ImageNet1k | | | CIFAR100 | | | Places365 | | |
|---|---|---|---|---|---|---|---|---|---|
| | $AC$ | $AA$ | $ACA$ | $AC$ | $AA$ | $ACA$ | $AC$ | $AA$ | $ACA$ |
| Encoder outputs | 0 | 21.63 | 0 | 0 | 18.83 | 0 | 0 | 8.38 | 0 |
| Softmax outputs (PSP) | 1 | 74.34 | 76.12 | 0.9 | 37.62 | 33.24 | 0.9 | 15.92 | 14.08 |
| Logits (LSP) | 0.7 | 60.68 | 37.61 | 0.4 | 45.60 | 17.58 | 0.2 | 21.36 | 4.06 |

**Advanced Network Architectures Results.** Tab. 2 presents the compatibility performance of various pretrained models on ImageNet1k (Russakovsky et al., 2015) to demonstrate that by simply using features derived from softmax outputs and logits models result to be compatible. In this case, we mimic a scenario in which a base model is updated five times every time with the same training data (aka ImageNet1k) but with a more advanced network architecture. Specifically, we started with AlexNet (Krizhevsky et al., 2012) than updated with ResNet50 (He et al., 2016), RegNetX_3.2GF (Radosavovic et al., 2020), ResNet152, and MaxViT_T (Tu et al., 2022). We evaluated how the compatible performance on the ImageNet1k dataset (closed-set results), the CIFAR100 dataset (open-set results) and Places365 (Zhou et al., 2017)(fine-grained open-set results) of PSP, LSP, and the baseline approach, in which features are derived from the softmax outputs, the logits, and the encoder outputs, respectively. In this case, none of the other compared methods can be applied as they require to train the updated model in order to optimize some additional loss functions or some specific modules, such as mapping functions or fixed classifiers.
In closed-set scenarios on ImageNet1k, softmax outputs (PSP) demonstrated the best compatibility performance due to their high alignment with target labels, leading to high $AC$ and $AA$. However, this alignment poses challenges in open-set scenarios like CIFAR100 and Places365, as highlighted by (Wang & Isola, 2020; Chen et al., 2022), where the alignment reduces transferability and overall open-set accuracy $AA$. Conversely, LSP, while offering lower accuracy in open sets compared to closed sets, achieves a higher overall accuracy, indicating a more robust and transferable representation. Additional performance details can be observed in the compatibility matrices of Appendix I for both the closed-set and open-set scenario. Appendix J presents the same experimental setup, focusing on the more expressive ViT (Dosovitskiy et al., 2020) architectures.
This experiments highlights not only the compatibility performance of PSP and LSP, but also how our methods can be applied directly to all the publicly available pretrained models without any additional modifications or training but simply through direct inference.

**Limitations.** PSP and LSP require that the ordering of classes does not change across model updates or, in the case of extended training classes, follows an extension within a nested dataset structure that does not alter the original sequence. Although this condition may seem restrictive, it can be mitigated if the permutations of the class order is known. A further limitation of our framework is that the representation size increases linearly with the number of classes (as can be seen from Eq. 2 and Eq. 6), as in the case of other methods (Biondi et al., 2023; Yang et al., 2022; Biondi et al., 2024) that leverage feature representations configured as Simplex. This results in feature vectors with high dimensions, which may lead to increased memory requirements. Although the representation benefits from top-$k$ sparsification for dimensionality reduction, as empirically confirmed (Appendix K), this promising and important aspect merits further investigation in future studies.

## 5 CONCLUSIONS

This paper showed that independently trained DNNs can be made compatible simply using feature representations derived from softmax outputs and logits. We demonstrated that these representations, configured as regular simplex, remain aligned across multiple model updates through projections. This results into stationary representations, which satisfy both the criteria of the definition of compatibility, recently shown to be not strictly feasible. Experiments confirmed the theoretical results, showing that our framework achieves state-of-the-art compatibility performance, particularly in scenarios with frequent model updates. Notably, our methodology achieves compatibility between publicly available pretrained models without requiring any additional training or adaptation.

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

## A PROOF FOR PROPOSITION 1

**Proposition.** *Assuming an increase in the number of classes to $C^t$ from $C^k$ at a new step $t$, the class prototypes of the newly added classes are orthogonal with respect to the prototypes from the previous step $k$.*

*Proof.* To simplify the notation in the following discussion, we will denote $C^k$ by $d$. Without loss of generality , we demonstrate this proposition for a step $t$ with an increment of one class relative to the previous step $k$, such that $C^t = C^k + 1 = d + 1$. This proof can be easily extended to model updates where more than one class are introduced.

Given a generic prototypes vector $\mathbf{v}_y^k$ of the Probability Simplex $\Delta^{d-1}$ in $\mathbb{R}^d$, where $d$ is the number of classes at task $k$ and $y$ is a generic class, its vector is defined by the direction from the center $\mathbf{o}^k = \frac{1}{d} \sum_{y=1}^d \mathbf{e}_y$ to the corresponding class vertex $\mathbf{e}_y$, i.e $\mathbf{v}_y^k = \mathbf{e}_y - \mathbf{o}^k = \mathbf{e}_y - \frac{1}{d}\mathbf{1}$. The $j$-th component $\mathbf{v}_{y,j}^k$ of $\mathbf{v}_y^k$ is defined as follow:

$$\mathbf{v}_{y,j}^k = \begin{cases} -\frac{1}{d} & \text{if } j \neq y \\ 1 - \frac{1}{d} & \text{if } j = y \end{cases}$$

When a new class is added to the training set, there is an orthogonal extension of the one-hot encoded label for the new added class. This causes a shift of the class prototypes as the evolving center of the new Probability Simplex $\Delta^d$ in $\mathbb{R}^{d+1}$ is defined as $\mathbf{o}^t = \frac{1}{d+1} \sum_{y=1}^{d+1} \mathbf{e}_y = \mathbf{e}_y - \frac{1}{d+1}\mathbf{1}$. The prototype vector of the new added class $d + 1$ is is then defined as $\mathbf{v}_{d+1}^k = \mathbf{e}_{d+1} - \mathbf{o}^t$.

To prove the orthogonality of the prototype of the new class with the class prototypes of the base model at step $k$, let compute the dot product between $\mathbf{v}_{d+1}^t$ and a generic prototype $\mathbf{v}_y^k \in \mathbb{R}^d$.

Let firstly compute the norm the two vectors. The norm of $\mathbf{v}_y^k$ is:

$$\|\mathbf{v}_y^k\| = \sqrt{\sum_{j=1}^d \left(v_{y,j}^k\right)^2} = \sqrt{\sum_{\substack{j=1 \\ j \neq y}}^d \left(v_{y,j}^k\right)^2 + \left(v_{y,y}^k\right)^2} = \sqrt{\sum_{\substack{j=1 \\ j \neq y}}^d \frac{1}{d^2} + \frac{(d-1)^2}{d^2}} =$$

$$= \sqrt{\frac{d-1}{d^2} + \frac{(d-1)^2}{d^2}} = \sqrt{\frac{d-1}{d}}$$

Thus, the normalized vector is

$$\frac{\mathbf{v}_y^k}{\|\mathbf{v}_y^k\|} = \frac{\mathbf{e}_y - \mathbf{o}^k}{\sqrt{\frac{d-1}{d}}} = \frac{\mathbf{e}_y - \frac{1}{d}\mathbf{1}}{\sqrt{\frac{d-1}{d}}}$$

The norm of $\mathbf{v}_{d+1}^t$, the prototype vector of the new added class $d + 1$, is

$$\|\mathbf{v}_{d+1}^t\| = \sqrt{\sum_{j=1}^{d+1} \left(v_{d+1,j}^t\right)^2} = \sqrt{\sum_{j=1}^d \left(v_{d+1,j}^t\right)^2 + \left(v_{d+1,d+1}^t\right)^2} =$$

$$= \sqrt{\sum_{j=1}^d \frac{1}{(d+1)^2} + \frac{d^2}{(d+1)^2}} = \sqrt{\frac{d}{(d+1)^2} + \frac{d^2}{(d+1)^2}} = \sqrt{\frac{d}{d+1}}$$

Thus, the normalized vector is

$$\frac{\mathbf{v}_{d+1}^t}{\|\mathbf{v}_{d+1}^t\|} = \frac{\mathbf{e}_{d+1} - \mathbf{o}^t}{\sqrt{\frac{d}{d+1}}} = \frac{\mathbf{e}_{d+1} - \frac{1}{d+1}\mathbf{1}}{\sqrt{\frac{d}{d+1}}}$$

The dot product between the two normalized vectors is calculated by padding the smaller one with one component of the vector with value equal to zero, to match their dimensions since $\mathbf{v}_y^k$ is in $\mathbb{R}^d$ and

$\mathbf{v}_{d+1}$ is in $\mathbb{R}^{d+1}$. This padding operation does not impact the normalization of the vector. Therefore, the padded vector $\mathbf{v}_y^k$ is:

$$\mathbf{v}_y^k = \left(\mathbf{e}_y - \frac{1}{d}\mathbf{1}, 0\right)$$

Then the dot product is given by:

$$\frac{\mathbf{v}_y^k}{\|\mathbf{v}_y^k\|} \cdot \frac{\mathbf{v}_{d+1}^t}{\|\mathbf{v}_{d+1}^t\|} = \sum_{\substack{j=1 \\ j \neq y}}^d \frac{v_{y,j}^k}{\|\mathbf{v}_y^k\|} \frac{v_{d+1,j}^t}{\|\mathbf{v}_{d+1}^t\|} + \frac{v_{y,y}^k}{\|\mathbf{v}_y^k\|} \frac{v_{d+1,y}^t}{\|\mathbf{v}_{d+1}^t\|} + \frac{v_{y,d+1}^k}{\|\mathbf{v}_y^k\|} \frac{v_{y,d+1}^t}{\|\mathbf{v}_{d+1}^t\|} \tag{9}$$

As $v_{y,d+1}^k = 0$, Eq. 9 simplifies to

$$\frac{\mathbf{v}_y^k}{\|\mathbf{v}_y^k\|} \cdot \frac{\mathbf{v}_{d+1}^t}{\|\mathbf{v}_{d+1}^t\|} = \frac{d-1}{d(d+1)\sqrt{\frac{d-1}{d}}\sqrt{\frac{d}{d+1}}} - \frac{d-1}{d(d+1)\sqrt{\frac{d-1}{d}}\sqrt{\frac{d}{d+1}}} = 0$$

It follows that, the dot product between a generic prototype vector $\mathbf{v}_y^k$ from the Probability Simplex $\Delta^{d-1}$ and the prototype vector of the newly added class $\mathbf{v}_{d+1}^t$ from the Probability Simplex $\Delta^d$ is:

$$\mathbf{v}_y^k \cdot \mathbf{v}_{d+1}^t = 0.$$

Thus, the two vectors are orthogonal. $\qquad\square$

## B  Proof for Proposition 2

**Proposition.** *Let $\mathbf{v}_y^t \in \mathbb{R}^{C^t}$ be the prototype vector for class $y$ within the Probability Simplex $\Delta^{C^t-1}$. Define its projection as $\mathbf{u}_y^t = \mathbf{P}_{t,k}\mathbf{v}_y^t$, where $\mathbf{P}_{t,k} = [\mathbf{V}^k|\mathbf{0}]$ with $\mathbf{V}^k = [\mathbf{v}_y^k]_{y=1}^{C^k} \in \mathbb{R}^{C^k \times C^k}$ and $\mathbf{0} \in \mathbb{R}^{(C^t-C^k)\times C^k}$ is a zero matrix. Then the resulting projected prototype $\mathbf{u}_y^t$ is aligned with the prototype $\mathbf{v}_y^k$ within $\Delta^{C^k-1}$.*

*Proof.* To simplify the notation in the following discussion, we will denote $C^k$ by $d$. Without loss of generality , we demonstrate this proposition for a step $t$ with an increment of one class relative to the previous step $k$, such that $C^t = C^k + 1 = d + 1$. This proof can be easily extended to model updates where more than one class are introduced.

Let $\mathbf{v}_y^t = \mathbf{e}_y - \mathbf{o}^t$ the prototype vector in $\mathbb{R}^{d+1}$ of $y$-th class following the direction from the center $\mathbf{o}^t = \frac{1}{d+1}\sum_{y=1}^{d+1}\mathbf{e}_y$ of the Probability Simplex $\Delta^d$ in $\mathbb{R}^{d+1}$ to its $y$-th vertex $\mathbf{e}_y$. Let $\mathbf{P}_{t,k} \in \mathbb{R}^{d \times (d+1)}$ be the projection matrix composed by $\mathbf{V}^k = [\mathbf{v}_y^k]_{y=1}^d \in \mathbb{R}^{d \times d}$, which is the matrix formed by the prototype vectors $\mathbf{v}_y^k$ as defined in Eq. 5 and the last column a zero vector, denoted as $\mathbf{0} \in \mathbb{R}^{d \times 1}$. This matrix $\mathbf{P}_{t,k}$ projects a vector $\mathbf{v}_y^t$ from $\mathbb{R}^{d+1}$ down to $\Delta^{d-1}$ in $\mathbb{R}^d$ centered in the orgin of the axes. The vector $\mathbf{v}_y^t$ can be explicitly express through its components as

$$\mathbf{v}_{y,j}^t = \begin{cases} -\frac{1}{d+1} & \text{if } j \neq y \\ 1 - \frac{1}{d+1} & \text{if } j = y \end{cases}$$

We define the projection of $\mathbf{v}_y^t$:

$$\mathbf{u}_y^t = \mathbf{P}_{t,k}\,\mathbf{v}_y^t$$

Since $\mathbf{P}_{t,k} = [\mathbf{V}^k\,|\,\mathbf{0}]$, when we apply the transformation $\mathbf{P}_{t,k}$ to $\mathbf{v}_y^t$, only the first $d$ components are considered (ignoring the $(d+1)$-th component) thanks to the vector of zeros $\mathbf{0}$. Therefore, the projected vector $\mathbf{u}_y^t \in \mathbb{R}^d$ is obtained by multiply $\mathbf{v}_y^t$ with $\mathbf{V}^k$:

$$\mathbf{u}_y^t = \mathbf{V}^k\mathbf{v}_y^t.$$

This operation can be decomposed into two parts, corresponding to the components of the vector $\mathbf{v}_y^t$, which is the $y$-th component of $\mathbf{u}^t$. The first one is the $y$-th element of the vector relative to vertex $\mathbf{e}_y$:

$$\mathbf{u}_{y,y}^t = \mathbf{v}_{y,y}^k \mathbf{v}_{y,y}^t + \sum_{\substack{j=1 \\ j \neq y}}^d \mathbf{v}_{y,j}^k \mathbf{v}_{y,j}^t = \frac{d-1}{d}\mathbf{v}_{y,y}^t - \frac{1}{d}\sum_{\substack{j=1 \\ j \neq y}}^d \mathbf{v}_{y,j}^t$$

where $\mathbf{v}_y^k$ are the columns of $\mathbf{V}^k$. Substituting the value of $\mathbf{v}_y^t$ in the equation, it can be simplified to

$$\mathbf{u}_{y,y}^t = \frac{d-1}{d}\frac{d}{d+1} - \frac{1}{d}(d-1)\left(-\frac{1}{d+1}\right) =$$

$$= \frac{d-1}{d+1} + \frac{d-1}{d(d+1)} = 1 - \frac{1}{d} = \mathbf{v}_{y,y}^k.$$

The second part is relative to the components of $\mathbf{v}_y^t$ where the value of $\mathbf{e}_y$ is equal to 0:

$$\mathbf{u}_{y,j}^t = \mathbf{v}_{y,j}^k \mathbf{v}_{y,y}^t + \mathbf{v}_{y,y}^k \mathbf{v}_{y,j}^t + \sum_{\substack{j=1 \\ j \neq y}}^{d-1} \mathbf{v}_{y,j}^k \mathbf{v}_{y,j}^t = -\frac{1}{d}\mathbf{v}_{y,y}^t + \frac{d-1}{d}\mathbf{v}_{y,j}^t - \frac{1}{d}\sum_{\substack{j=1 \\ j \neq y}}^{d-1} \mathbf{v}_{y,j}^t =$$

$$= -\frac{1}{d}\frac{d}{d+1} + \frac{d-1}{d}\left(-\frac{1}{d+1}\right) - \frac{1}{d}(d-2)\left(-\frac{1}{d+1}\right) = -\frac{1}{d} = \mathbf{v}_{y,j}^k.$$

Thus, each component of the projected vector $\mathbf{u}_y^t$ and $\mathbf{v}_y^k$ are equal, then the two vectors are equivalent:

$$\mathbf{u}_y^t = \mathbf{P}_{t,k}\,\mathbf{v}_y^t = \mathbf{v}_y^k.$$

It follows that $\mathbf{u}_y^t \in \mathbb{R}^d$, the projection of $\mathbf{v}_y^t \in \mathbb{R}^{d+1}$, is aligned with the prototype vector $\mathbf{v}_y^k \in \mathbb{R}^d$ through a projection matrix $\mathbf{P}_{t,k}$. $\qquad\square$

## C  PROOF FOR THEOREM 1

**Theorem** (*Probability Simplex Compatibility Theorem*). *Assuming the number of classes changes from $C^k$ at step $k$ to $C^t$ at step $t$, where $C^t \geq C^k$, the normalized softmax features of two models, independently trained at these respective steps, are compatible as formulated in Def. 1.*

*Proof.* In light of Proposition 2, the projected normalized softmax outputs of two independently trained models, one with an increased or equal class count $C^t$, can be considered embedded within a common hypersphere $S^{C^k-1}$.

To simplify the notation in the following discussion, we will denote the number of classes $C^k$ by $d$. Let now consider $\mathbf{X}, \mathbf{Y} \in S^{d-1}$ be two[1] independent random vectors drawn from the von Mises-Fisher (vMF) distributions with parameters $(\boldsymbol{\mu}_1, \kappa_1)$ and $(\boldsymbol{\mu}_2, \kappa_2)$, respectively. The vMF distribution on the unit hypersphere $S^{d-1}$ is characterized by the mean direction $\boldsymbol{\mu} \in S^{d-1}$, a unit vector indicating the direction around which the data is concentrated, and the concentration parameter $\kappa \geq 0$, a non-negative scalar that determines the concentration of the distribution around $\boldsymbol{\mu}$. Larger values of $\kappa$ correspond to higher concentration around the mean direction.

The probability density function (pdf) of the vMF distribution is given by:

$$f(\mathbf{x}; \boldsymbol{\mu}, \kappa) = c_d(\kappa)\exp\left(\kappa\boldsymbol{\mu}^\top \mathbf{x}\right), \quad \mathbf{x} \in S^{d-1},$$

where $c_d(\kappa)$ is the normalization constant ensuring that the pdf integrates to one over the unit sphere. The normalization constant is defined as:

$$c_d(\kappa) = \frac{\kappa^{d/2-1}}{(2\pi)^{d/2}I_{d/2-1}(\kappa)},$$

---

[1]This assumption of two random vectors is because we focus only on a single pairwise class interaction, since all other interactions are symmetrically similar due to the simplex symmetry and therefore do not change.

and $I_v(\kappa)$ is the modified Bessel function of the first kind of order $v$:

$$I_v(\kappa) = \left(\frac{\kappa}{2}\right)^v \sum_{j=0}^{\infty} \frac{\left(\frac{\kappa^2}{4}\right)^j}{j!\,\Gamma(v+j+1)}.$$

The goal is to compute the expected value of the cosine distance $1 - \cos\theta$, where $\theta$ is the angle between $\mathbf{X}$ and $\mathbf{Y}$: $\theta = \arccos\left(\mathbf{X}^\top \mathbf{Y}\right)$. We aim to derive $\mathbb{E}[1 - \cos\theta]$ as a function of the angle $\alpha$ between $\boldsymbol{\mu}_1$ and $\boldsymbol{\mu}_2$, and the concentration parameters $\kappa_1$ and $\kappa_2$.

To compute $\mathbb{E}[1 - \cos\theta]$, we start by observing that:

$$\mathbb{E}[1 - \cos\theta] = 1 - \mathbb{E}\left[\mathbf{X}^\top \mathbf{Y}\right]. \tag{10}$$

Assuming $\mathbf{X}$ and $\mathbf{Y}$ independent[2], the expectation of their inner product can be expressed as the product of their expectations: $\mathbb{E}\left[\mathbf{X}^\top \mathbf{Y}\right] = \mathbb{E}\left[\mathbf{X}\right]^\top \mathbb{E}\left[\mathbf{Y}\right]$. The expected values of $\mathbf{X}$ and $\mathbf{Y}$ are given by:

$$\mathbb{E}\left[\mathbf{X}\right] = m_d\left(\kappa_1\right) \boldsymbol{\mu}_1,$$
$$\mathbb{E}\left[\mathbf{Y}\right] = m_d\left(\kappa_2\right) \boldsymbol{\mu}_2,$$

where the *mean resultant length* $m_d(\kappa)$ is defined as:

$$m_d(\kappa) = \frac{I_{d/2}(\kappa)}{I_{d/2-1}(\kappa)}.$$

This quantity represents the expected value of the cosine of the angle between a random vector $\mathbf{X}$ drawn from the vMF distribution and the mean direction $\boldsymbol{\mu}$:

$$\mathbb{E}\left[\boldsymbol{\mu}^\top \mathbf{X}\right] = m_d(\kappa).$$

Substituting the expressions for $\mathbb{E}\left[\mathbf{X}\right]$ and $\mathbb{E}\left[\mathbf{Y}\right]$, we obtain:

$$\mathbb{E}\left[\mathbf{X}^\top \mathbf{Y}\right] = \left(m_d\left(\kappa_1\right) \boldsymbol{\mu}_1\right)^\top \left(m_d\left(\kappa_2\right) \boldsymbol{\mu}_2\right) = m_d\left(\kappa_1\right) m_d\left(\kappa_2\right) \boldsymbol{\mu}_1^\top \boldsymbol{\mu}_2.$$

The inner product $\boldsymbol{\mu}_1^\top \boldsymbol{\mu}_2$ equals $\cos\alpha$, where $\alpha$ is the angle between the mean directions $\boldsymbol{\mu}_1$ and $\boldsymbol{\mu}_2$:

$$\cos\alpha = \boldsymbol{\mu}_1^\top \boldsymbol{\mu}_2.$$

Therefore, we have:

$$\mathbb{E}\left[\mathbf{X}^\top \mathbf{Y}\right] = m_d\left(\kappa_1\right) m_d\left(\kappa_2\right) \cos\alpha.$$

Substituting back into Eq. 10, we find:

$$\mathbb{E}[1 - \cos\theta] = 1 - m_d\left(\kappa_1\right) m_d\left(\kappa_2\right) \cos\alpha. \tag{11}$$

This expression relates the expected value of the cosine distance to the angle $\alpha$ between the mean directions and the concentration parameters $\kappa_1$ and $\kappa_2$. It is important to note that this computation does not require any approximation, as we have directly calculated $\mathbb{E}[1 - \cos\theta]$ using the properties of the von Mises-Fisher distribution and the independence of $\mathbf{X}$ and $\mathbf{Y}$.

Since the mean resultant length $m_d(\kappa)$ is a strictly increasing function of $\kappa$ for $\kappa > 0$, it follows that:

$$m_d(\eta\kappa) > m_d(\kappa), \ \forall \eta > 1. \tag{12}$$

This implies that higher concentration parameters result in random vectors that are more tightly clustered around their mean directions, leading to higher values of $m_d(\kappa)$.

---

[2]This assumption is based on the hypothesis that the deep neural network models show sufficient expressiveness to learn any tasks, as presented within the Unconstrained Feature Model and Layered Peeled Model frameworks Mixon et al. (2022); Fang et al. (2021), respectively. The assumption is supported by the Neural Collapse phenomenon Papyan et al. (2020), observed in different networks and datasets, including two-layer neural networks with independent input feature. This equivalence supports the notion that typical neural network architectures have sufficient capacity to learn features as statistically independent random variables.

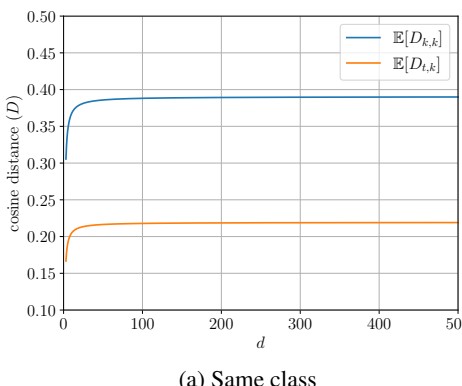 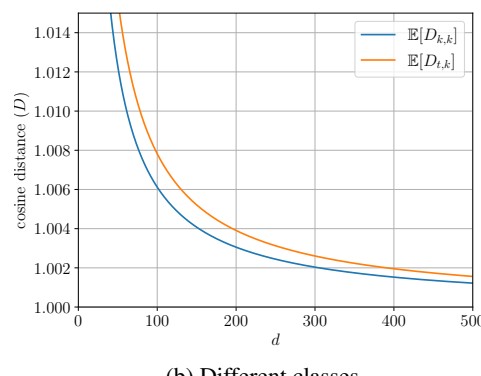

(a) Same class

(b) Different classes

Figure 5: Analytic expression of expected cosine distances between PSP feature representation from two different random variables, each following a distinct von Mises-Fisher distribution, across various dimensions of the representation space during two learning phases. These phases are identified by the time step $k$ to refer to the base model and $t$ to refer to the updated model. Both $\mathbb{E}[D_{t,k}]$ and $\mathbb{E}[D_{k,k}]$ are analyzed, where $\mathbb{E}[D_{t,k}]$ is the expected cosine distance between a more concentrated distribution at step $t$ respect to the distribution at step $k$. (a) In the case of the same class, the value of $\mathbb{E}[D_{t,k}]$ remains consistently lower than $\mathbb{E}[D_{k,k}]$, which on average satisfies the condition of Eq. 1a. (b) In the case of a different class, the value of $\mathbb{E}[D_{t,k}]$ remains consistently higher than $\mathbb{E}[D_{k,k}]$, which on average satisfies the condition of Eq. 1b.

According to this analysis, an increase in concentration parameters leads to a decrease in $\mathbb{E}[1 - \cos\theta]$ when the mean directions are less than $\pi/2$ and an increase in $\mathbb{E}[1 - \cos\theta]$ when they exceed $\pi/2$. This observation highlights the influence of directional concentration and angular displacement on distribution behavior.

We assume that after an update the newly learned information improves the model's discrimination capability. Consequently, this leads to a greater concentration in the von Mises-Fisher (vMF) distribution of the updated model's class features, consistent with the condition $\eta > 1$ as specified in Eq. 12. According to this, the application of Eq. 11 between a base and an updated model consistently verifies the compatibility as in Def. 1. Fig. 5 shows this behavior while varying the classes $d$. If there is not new information to improves the model's discrimination capability, this leads to equal concentration in the von Mises-Fisher (vMF) distribution of the updated model's class features, corresponding to the condition $\eta = 1$. As the mean resultant length $m_d(\kappa)$ does not change with the model update, the inequalities of Def. 1 are satisfied (it correspond to the trivial solution discussed by Shen et al. (2020)). □

## D  PROOF FOR PROPOSITION 3

**Proposition.** *As the softmax outputs approach the vertices of the Probability Simplex, the corresponding logits vectors assume a simplex configuration, with class prototypes aligning to the vectors specified in Eq. 5.*

*Proof.* The cross-entropy loss function for a single sample $\mathbf{x}$ with class label $y$ (one-hot encoded as $\mathbf{y}$) and its softmax probabilities $\sigma(\mathbf{z})$ is:

$$\mathcal{L}(\sigma(\mathbf{z}), \mathbf{y}) = -\log(\sigma(\mathbf{z})_y) = -\log\left(\frac{e^{\mathbf{z}_y}}{\sum_{j=1}^{C} e^{\mathbf{z}_j}}\right)$$

where $\sigma(\mathbf{z})_y$, $\mathbf{z}_y$ indicates the $y$-th component of the softmax output and logits vector, being $\sigma(\cdot)$ the softmax function. The gradient of the cross-entropy loss with respect to the $j$-th component of the logits vector $\mathbf{z}$ is:

$$\frac{\partial \mathcal{L}}{\partial \mathbf{z}_j} = \sigma(\mathbf{z})_j - \mathbf{y}_j = \begin{cases} \sigma(\mathbf{z})_j - 1 & \text{if } j = y \\ \sigma(\mathbf{z})_j & \text{if } j \neq y \end{cases}$$

being $\mathbf{y}_j$ the $j$-th component of the one-hot encoded labels $\mathbf{y}$.

Using gradient descent, the update rule for the $j$-th component $\mathbf{z}_j$ of the logits vector $\mathbf{z}$ is:

$$\mathbf{z}_j \leftarrow \mathbf{z}_j - \eta \frac{\partial \mathcal{L}}{\partial \mathbf{z}_j}$$

where $\eta$ is the learning rate. Substituting the gradients, we get:

$$\mathbf{z}_j \leftarrow \mathbf{z}_j - \eta\big(\sigma(\mathbf{z})_j - \mathbf{y}_j\big) = \begin{cases} \mathbf{z}_j - \eta(\sigma(\mathbf{z})_j - 1) & \text{if } j = y \\ \mathbf{z}_j - \eta\,\sigma(\mathbf{z})_j & \text{if } j \neq y \end{cases}$$

Since $\sigma(\mathbf{z})_y$ is optimized to be 1 and $\sigma(\mathbf{z})_j$ with $j \neq y$ to be 0, the gradient updates increase the logit value for the correct class label $y$ while decreasing the logits for the other classes $j \neq y$, pushing the logits vector $\mathbf{z}$ towards the direction of the class prototype vector $\mathbf{v}_y = \mathbf{y} - \mathbf{o} = \mathbf{e}_y - \mathbf{o}$.

If we reasonable to assume that, at the beginning of training, logits are distributed near the center of the origin of the axes, doing a gradient step update, we get

$$\mathbf{z}_j = \begin{cases} 0 + \eta\left(1 - \frac{1}{C}\right) & = \eta\, v_y & \text{if } j = y \\ 0 + \eta\left(-\frac{1}{C}\right) & = \eta\, v_j & \text{if } j \neq y \end{cases}$$

being $\mathbf{v}_y = [v_1, v_2, \ldots, v_y, \ldots, v_C]$.

This implies that, at each update, logits align towards the direction of respective class prototype. $\quad\square$

# E  PSP AND LSP CONVERGENCE TO SIMPLEX CONFIGURATION

In this section, we leverage the Neural Collapse (NC) hypothesis (Papyan et al., 2020) to show that, in the terminal phase of training, logits are configured in a simplex and present a wider spread in feature distributions with respect to softmax outputs. This evidence holds for several datasets and neural network architectures.

Let $\mathbf{h}_{i,y}$ a feature extracted by a model in response to an image $\mathbf{x}_i$ of class $y$, $\boldsymbol{\mu}_y = \text{avg}\{\mathbf{h}_{i,y}\}$ the feature class-mean and $\boldsymbol{\mu}_G = \text{avg}\{\boldsymbol{\mu}_y \text{ for } y = 1, 2, \ldots, C\}$ the mean of class-means. It follows that:

**(NC1) Variability collapse:** The within-class covariance of features collapse to zero:

$$\text{avg}\{\mathbf{h}_{i,y} - \boldsymbol{\mu}_y\} \rightarrow \mathbf{0}$$

**(NC2) Equinorm and Equidistance of Features Class-means:** The class means of features tend to form a simplex. A simplex is a symmetric structure whose vertices lie on a hyper-sphere (i.e., they have same norm) and are placed at the maximum possible distance from each other. Being $y' \neq y$ a generic other class, it holds that:

$$\big|\|\boldsymbol{\mu}_y - \boldsymbol{\mu}_G\|_2 - \|\boldsymbol{\mu}_{y'} - \boldsymbol{\mu}_G\|_2\big| \rightarrow 0 \quad \forall\, y, y'$$

$$\langle \tilde{\boldsymbol{\mu}}_y, \tilde{\boldsymbol{\mu}}_{y'} \rangle \rightarrow \frac{C}{C-1} \delta_{y,y'} - \frac{1}{C-1} \quad \forall\, y, y'$$

**(NC3) Convergence to self-duality:** Class-means and classifiers weights converge to each other.

$$\left\| \frac{\mathbf{W}^\top}{\|\mathbf{W}\|_F} - \frac{\mathbf{M}}{\|\mathbf{M}\|_F} \right\|_F \rightarrow 0$$

In the case of softmax outputs and logits, NC3 reduces to evaluate the distance of features with respect to $\mathbf{V}^k = [\mathbf{v}_y^k]_{y=1}^{C^k} \in \mathbb{R}^{C^k \times C^k}$, that is the matrix obtained by stacking the prototype vectors $\mathbf{v}_y^k$ defined in Eq. 5 for each class $y$, i.e.,

$$\left\| \frac{\mathbf{V}^k}{\|\mathbf{V}^k\|_F} - \frac{\mathbf{M}}{\|\mathbf{M}\|_F} \right\|_F \rightarrow 0$$

**(NC4): Simplification to NCC:** When a feature point $\mathbf{h}^\star$ has to be classified, the decision rule reduces to choose the nearest class-means.

$$\arg\max_{y'} \mathbf{h}^\star \rightarrow \arg\min_{y'} \|\mathbf{h}^\star - \boldsymbol{\mu}_{y'}\|_2$$

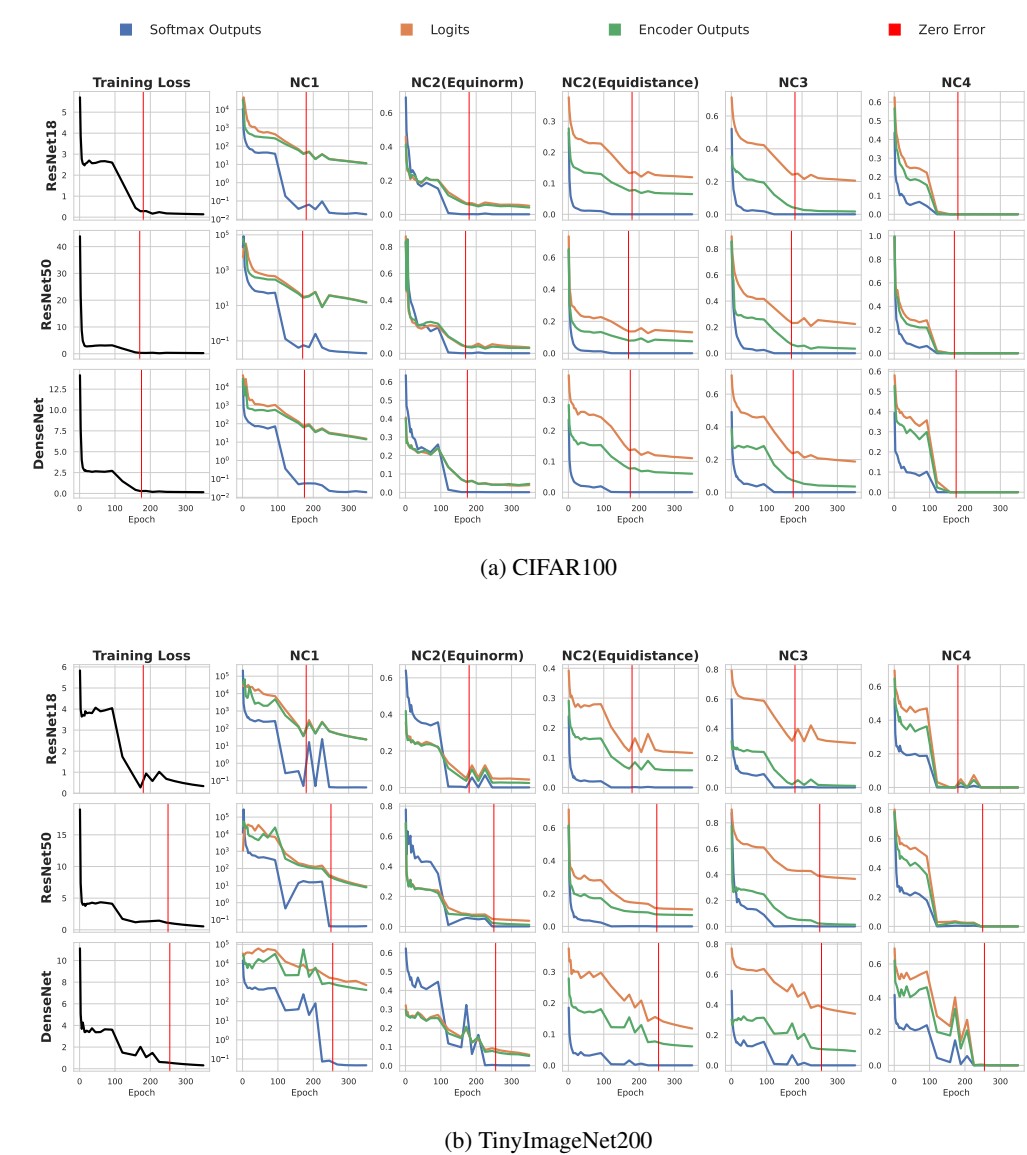

(a) CIFAR100

(b) TinyImageNet200

Figure 6: Neural Collapse hypothesis evaluated for softmax outputs (blue lines), logits (orange lines), and encoder outputs (green lines) across different network architecture and datasets, showing logits presenting a reduced alignment to the simplex configuration with a wider spread in feature distribution. This can be beneficial for a robust and transferable representation.

where $\tilde{\boldsymbol{\mu}}_y = (\boldsymbol{\mu}_y - \boldsymbol{\mu}_G)/\|\boldsymbol{\mu}_y - \boldsymbol{\mu}_G\|_2$ are the renormalized the class-means, $\mathbf{M} = [\boldsymbol{\mu}_y - \boldsymbol{\mu}_G]_{y=1}^C$ is the matrix obtained by stacking the class-means into columns, $\mathbf{W}$ is the classifier weights matrix, and $\delta_{y,y'}$ is the Kronecker delta symbol.

Fig. 6 presents NC hypothesis plots for softmax outputs (blue), logits (orange), and encoder outputs (green) across different network architectures (ResNet18, ResNet50, and DenseNet) and datasets (CIFAR100 and TinyImageNet200). The figure shows consistent trends across various neural networks and datasets for softmax outputs, logits, and encoder outputs. This behavior suggests that logits may balance the trade-off between alignment and generalization better than softmax outputs and encoder outputs, as described in Chen et al. (2022). The NC2 and NC3 curves show that logits are the less collapsed onto the simplex vertices—indicating reduced alignment—with a broader spread feature distribution—indicating better generalization, as evidenced by the NC1 values. This because although logits present a spread comparable to the encoder outputs one, their alignment towards the

fixed simplex reference $\mathbf{V}^k$ is more complex to obtain. This does not hold for softmax outputs as they converge towards the vertices of the Probability Simplex every time the training loss approaches zero.

## F  PSEUDO-CODE

As shown in Alg. 1, the pseudocode outlines the computation of PSP or LSP feature vectors. It accepts inputs of class counts from base and updated models, and the softmax or logit outputs from updated model queries. The output is a normalized feature vector $\mathbf{h}^t$. The procedure "GetPSPFeatures" constructs a transformation matrix used to transform and then normalize the feature vector.

---

**Algorithm 1** Compute PSP (LSP) Feature Representation

---

1: **Input:** Number of classes $C^k$, $C^t$ of the base and updated models respectively, and the softmax (for PSP) or logits (for LSP) output vector $\mathbf{f}^t$ of some query input data $\mathbf{x}$ obtained with the updated model at step $t$.
2: **Output:** The PSP (LSP) feature representation vector denoted as $\mathbf{h}^t$.
3: **procedure** GETPSPFEATURES($\mathbf{f}^t, C^t, C^k$)
4:      $\mathbf{P}_{k,k} = \mathbf{I}_{C^k} - \frac{1}{C^k}\mathbf{J}_{C^k}$
5:      $\mathbf{0} = \text{ZeroMatrix}(C^t - C^k, C^k)$
6:      $\mathbf{P}_{t,k} = [\mathbf{P}_{k,k} \mid \mathbf{0}]$
7:      $\mathbf{h}^t = \mathbf{P}_{t,k}\mathbf{f}^t$
8:      $\mathbf{h}^t = \frac{\mathbf{h}^t}{\|\mathbf{h}^t\|}$
9:      **return** $\mathbf{h}^t$
10: **end procedure**

---

## G  IMPLEMENTATION DETAILS

In the following we report the implementations details we used in our experiments. All the values reported in Sec. 4 are obtained with the same training hyperparameters on a Nvidia Quadro A6000 GPU with 24GB and two Nvidia A100 GPUs, each with 40GB.

**CIFAR100** (Krizhevsky, 2009).   Images are 32×32. A ResNet18 architecture was used with the following hyper-parameters for training: number of epochs 120; batch size 128; SGD optimizer with learning rate that starts from 0.1 and is divided by 10 after 80 and 100 epochs. SGD momentum 0.9 and weight decay is set to $5 \cdot 10^{-4}$. A temperature factor of 12 has been used in the cross-entropy loss to scale logits vectors during training. Training images were subjected to random cropping, horizontal flipping, and tensor normalization. CIFAR100 classes are divided to have at each step an equal number of new classes, i.e., $|\mathcal{X}_t| = 100/T$ for $t = 1, 2, \ldots, T$.

**TinyImageNet200** (Le & Yang, 2015).   Images are resized to 64×64. A ResNet18 architecture was used with the following hyper-parameters for training: number of epochs 90; batch size 256; SGD optimizer with learning rate that starts from 0.1 and is divided by 10 after 50 and 70 epochs. SGD momentum 0.9 and weight decay is set to $5 \cdot 10^{-4}$. A temperature factor of 12 has been used in the cross-entropy loss to scale logits vectors during training. Training images were subjected to random cropping, horizontal flipping, and tensor normalization. TinyImageNet200 classes are divided to have at each step an equal number of new classes, i.e., $|\mathcal{X}_t| = 200/T$ for $t = 1, 2, \ldots, T$.

**ImageNet1K** (Russakovsky et al., 2015).   Images are resized to 224×224. A ResNet50 architecture was used with the following hyper-parameters for training: number of epochs 90; batch size 1536; SGD optimizer with learning rate that starts from 0.1 and is divided by 10 after 30 and 60 epochs. SGD momentum 0.9 and weight decay is set to $1 \cdot 10^{-4}$. Training images were subjected to random cropping, horizontal flipping, color jitter, and tensor normalization. ImageNet1k classes are divided to have at each step an equal number of new classes, i.e., $|\mathcal{X}_t| = 1000/T$ for $t = 1, 2, \ldots, T$.

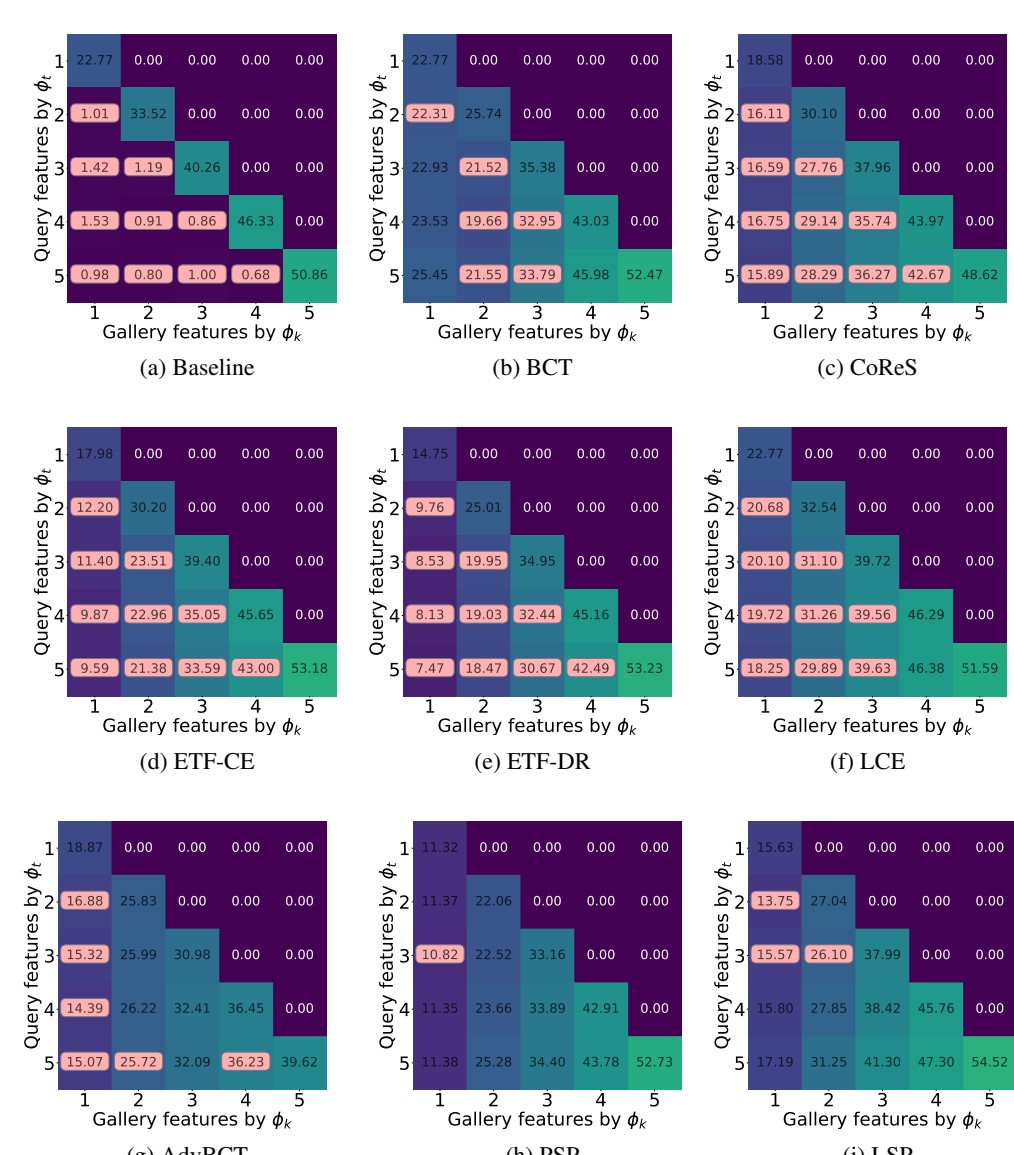

Figure 7: Compatibility matrices for PSP, LSP, and all other methods for the CIFAR100 5-step update setting, presented in Tab. 1a. The reported values are Recall@1 between query features $\Phi_{(\cdot)}^{\mathcal{Q}}$ and gallery features $\Phi_{(\cdot)}^{\mathcal{G}}$. Entries that do not satisfy the compatibility condition (Eq. 7) are highlighted with a light-red background.

**Google Landmarks v2** (Weyand et al., 2020). Images are resized to 224×224. A ResNet18 architecture, pretrained with ImageNet1k, was used with the following hyper-parameters for training: number of epochs 30; batch size 512; SGD optimizer with learning rate that starts from 0.1 and is divided by 10 after 5, 10 and 20 epochs. SGD momentum 0.9 and weight decay is set to $5 \cdot 10^{-4}$. Training images were subjected to random cropping and tensor normalization. Initial step of Google Landmarks v2 has 24393 classes, the others have same number of classes where the remaining classes are divided to have at each step an equal number of new classes, i.e., $|\mathcal{X}_t| = (81313 - 24393)/T$ for $t = 2, 3, \ldots, T$.

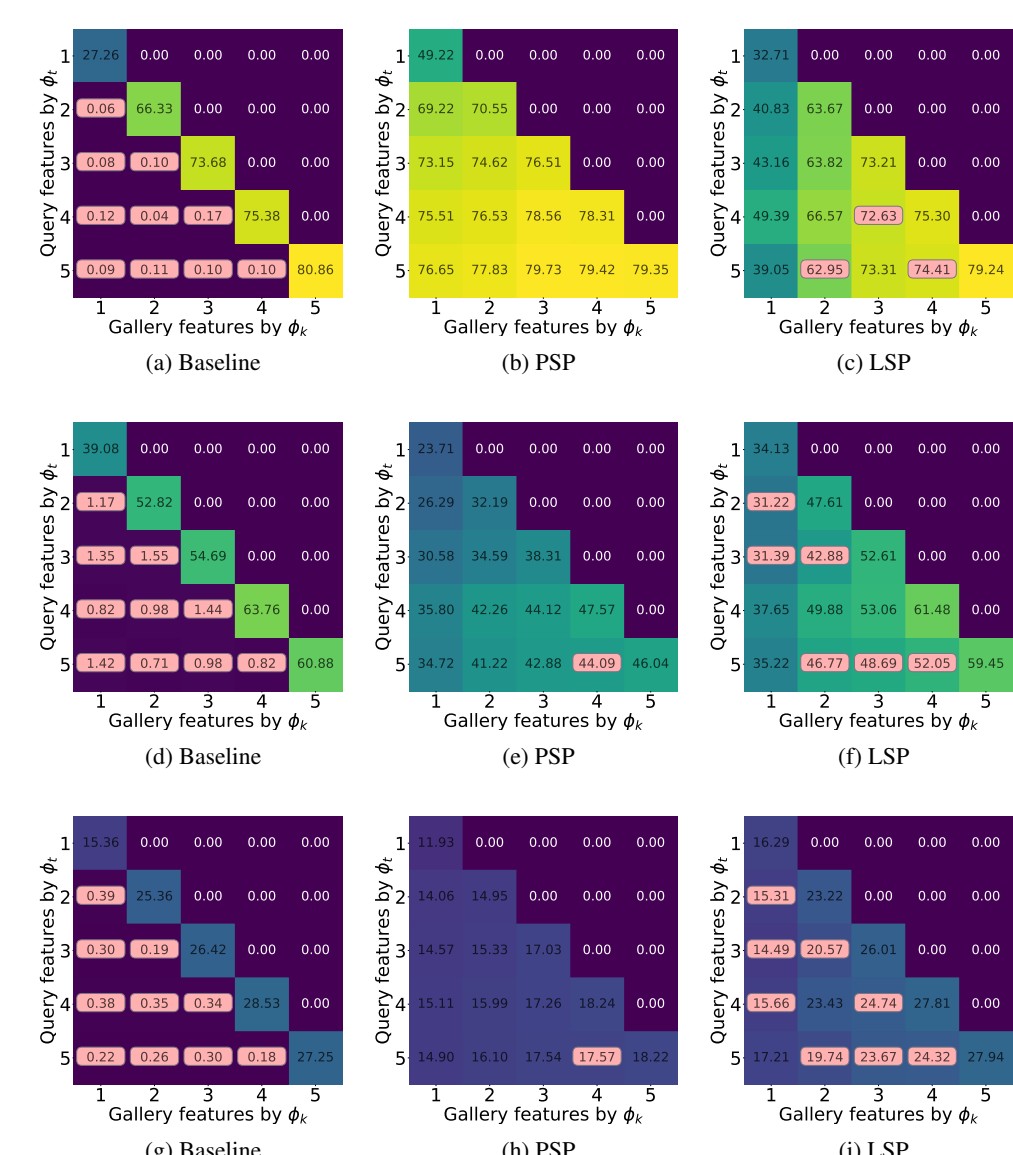

Figure 8: Compatibility Matrices of PSP, LSP, and baseline approach for 5 step in the case of advanced network architectures (AlexNet, ResNet50, RegNetX_3.2GF, ResNet152, and MaxViT_T). All the models were pretrained on ImageNet1k. (a), (b), (c) report the closed-set Recall@1 on the ImageNet1k dataset; (d), (e), (f) the open-set Recall@1 on the CIFAR100 dataset; (g), (h), (i) the fine-grained open-set Recall@1 on the Places365 dataset. Entries that do not satisfy compatibility Eq. 7 are highlighted with light-red background.

## H    EXTENDED CLASS: COMPATIBILITY MATRICES

To provide a more in depth insight of Tab. 1 into how performance varies across update steps for both self-tests and cross-tests, Fig.7 presents the compatibility matrices for the CIFAR100 5-step update scenario for each method. The main diagonals of the compatibility matrices (self-tests) capture the performance improvements obtained by adding new classes in the training, as they evaluate the Recall@1 of the model when both query and gallery features are extracted using the same model. The off-diagonal values represent the cross-test values, namely the performance of using the newer model as query-set feature extractor and the older one to obtain features of the gallery-set.

Table 3: Experimental results on ImageNet1k, CIFAR100 and Places365 in the case of advanced network architectures (ViT-B-32, ViT-B-16, and ViT-L-16), evaluated using the $AC$, $AA$, and $ACA$ metrics, with Recall@1 as the performance metric ($M$). Models are pretrained on ImageNet1k. Dark blue numbers indicate the highest values, while light blue the second-highest values.

| | ImageNet1k | | | CIFAR100 | | | Places365 | | |
|---|---|---|---|---|---|---|---|---|---|
| Features derived from | $AC$ | $AA$ | $ACA$ | $AC$ | $AA$ | $ACA$ | $AC$ | $AA$ | $ACA$ |
| Encoder outputs | 0 | 38.62 | 0 | 0 | 34.94 | 0 | 0 | 15.10 | 0 |
| Softmax outputs (PSP) | 1 | 78.93 | 80.52 | 0.66 | 52.60 | 37.94 | 1 | 18.31 | 17.60 |
| Logits (LSP) | 0.66 | 71.82 | 45.90 | 0.23 | 61.78 | 20.29 | 0.23 | 25.70 | 7.988 |

## I  ADVANCED NETWORK ARCHITECTURES: COMPATIBILITY MATRICES

Fig. 8 presents the compatibility matrices for PSP, LSP, and the baseline approach over 5 steps in the scenario of advanced network architectures. We utilized publicly available ImageNet1k pretrained models[3] to simulate a scenario where, starting from AlexNet as the base model, the network architectures are sequentially updated to ones with increasing expressiveness (ResNet50, RegNetX_3.2GF, ResNet152, and MaxViT_T). The matrices show that using softmax outputs as features (PSP) consistently achieves the highest number of compatible representations, both for ImageNet1k (Fig. 8a, Fig. 8b, Fig. 8c), CIFAR100 (Fig. 8d, Fig. 8e, and Fig. 8f), and Places365 (Fig. 8g, Fig. 8h, and Fig. 8i). While LSP is less compatible than PSP, it reports comparable Recall@1 values in the closed-set (ImageNet1k) scenario and higher values in the open-set (CIFAR100 and Places365) scenario. Notably, there is no significant drop in performance also compared to the baseline approach, which uses the standard practice of extracting features from the output of the encoder of the model. This confirm that LSP better manage the trade-off between alignment and generalization than both PSP and the baseline approach, as demonstrated in Appendix E.

## J  ADVANCED NETWORK ARCHITECTURES WITH VIT

Tab. 3 summarizes the compatibility performance of our method using ViT (Dosovitskiy et al., 2020) as the network architecture. The results demonstrate that features derived from softmax outputs (PSP) and logits (LSP) improve the compatibility as ViT's expressive power increases. These results align with those discussed in Section 4 for other network architectures.

To provide a more detailed view of performance in this context, Fig. 9 shows the compatibility matrices for PSP, LSP, and the baseline approach. We evaluate the compatibility of three ViT models: first, ViT-B-32, trained from scratch on ImageNet1k; followed by ViT-B-16, also trained from scratch on ImageNet1k; and finally, ViT-L-16, which was fine-tuned on ImageNet1k after pretraining via self-supervised learning (Singh et al., 2022). The matrices reveal that PSP achieves the highest compatibility score across datasets, in both the closed-set setting with ImageNet1k (Fig. 9a, Fig. 9b, Fig. 9c) and in the open-set one with CIFAR100 (Fig. 9d, Fig. 9e, and Fig. 9f) and Places365 (Fig. 9g, Fig. 9h, and Fig. 9i).

## K  DIMENSIONALITY REDUCTION WITH TOP-$k$ SPARSIFICATION

The size of PSP and LSP representations increases linearly with the class count, which can be challenging when using datasets with a large data diversity. This situation is similar to challenges observed in other methods (Biondi et al., 2023; Yang et al., 2022; Biondi et al., 2024) that utilize feature representations based on the regular simplex. However, in the case of PSP representations, top-$k$ sparsification (Lin et al., 2018; Zheng et al., 2023) can be employed for dimension reduction, motivated by the established principles of softmax and top-$k$ operations. Indeed, top-$k$ can be justified by how PSP can selectively ignores components of the softmax output vectors with minimal activation (i.e., low probability outputs). Thus, in an open-set scenario, these top-$k$ components might represent the objects most similar to the input class, highlighted by those classes showing some activation in the softmax outputs. From a theoretical perspective, Lapin et al. (2016) demonstrates that softmax

---

[3]`https://pytorch.org/vision/stable/models.html`

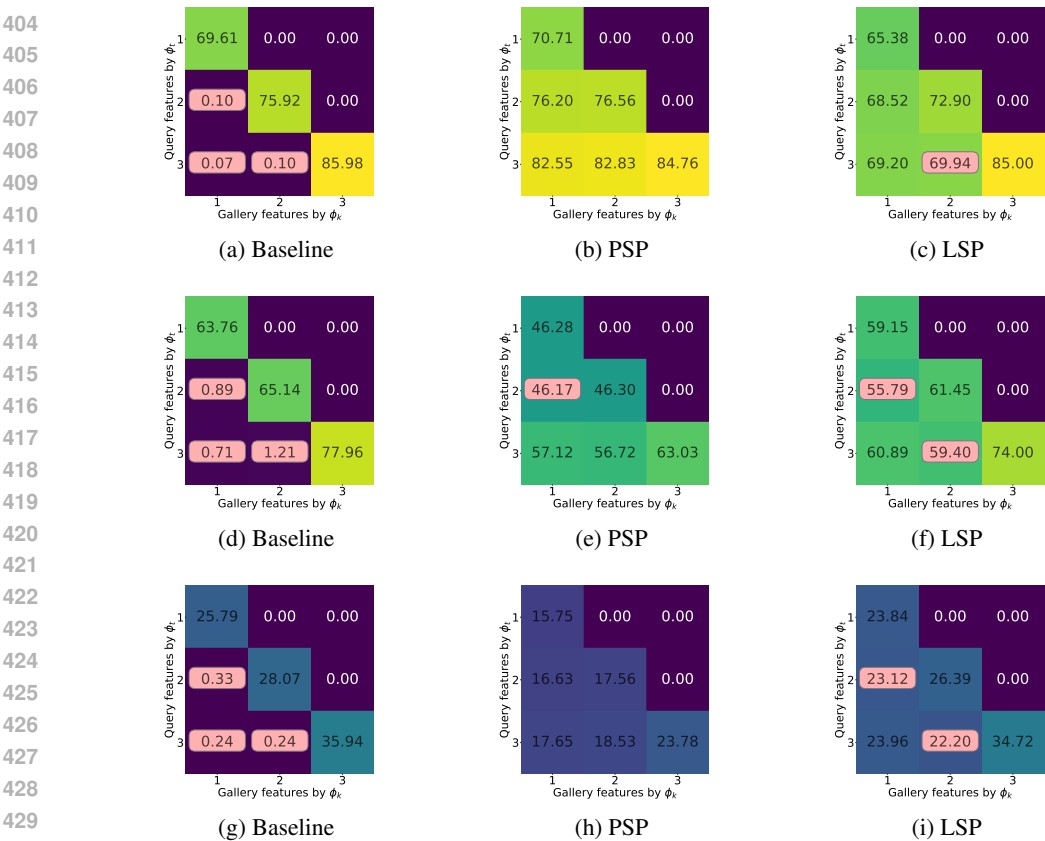

Figure 9: Compatibility Matrices of PSP, LSP, and baseline approach for 3 step in the case of advanced ViT network architectures (ViT-B-32, ViT-B-16, and ViT-L-16). All the models were pretrained on ImageNet1k. (a), (b), (c) report the closed-set Recall@1 on the ImageNet1k dataset; (d), (e), (f) the open-set Recall@1 on the CIFAR100 dataset; (g), (h), (i) the open-set Recall@1 on the Places365 dataset. Entries that do not satisfy compatibility Eq. 7 are highlighted with light-red background.

loss yields competitive top-$k$ performance for *all* values of $k$ simultaneously. This indicates that it preserves the ranking of the probabilities, which can contribute to keep the order of the activated components across different dimensions when they are transformed by our PSP method. Accordingly, top-$k$ sparsification seems to be well-suited for dimensionality reduction by setting all entries of a PSP feature representation vector to zero except for the top-$k$. Although LSP lacks similar theoretical support because logits are not probabilistically normalized, our alignment proof with respect to PSP, and the fact that both share similar hyperspherical geometry, suggests that they might behave similarly. In Tab. 4, we report results for PSP and LSP, where top-$k$ dimensional reduction has been applied to reduce features to the top-128 dimensions. For BCT and LCE, the feature size is reduced to the same 128 dimensions via a fully-connected layer, facilitating a direct comparison of the two methods under the same feature dimensions. These models were trained on the TinyImageNet200 training set and evaluated on the TinyImageNet200 validation set using Recall@1 at 2, 5, 20, and 50 update steps. Both PSP and LSP show slightly improved compatibility ($AC$) and accuracy ($AA$) performance in all cases compared to the results in Tab. 1b, where no dimension reduction was applied. Conversely, all compared methods exhibit a decrease in compatibility and accuracy due to the constrained feature space. Specifically, for BCT and LCE, performance is comparable to that in Tab. 1b during infrequent updates (2 and 5 steps) but declines significantly with more frequent updates.

Table 4: Comparison of Methods with Same Feature Length. Evaluation is performed on Tiny-ImageNet200 using the Recall@1 metric ($M$) for $AC$, $AA$, and $ACA$. PSP and LSP employ a top-$k$ feature dimension reduction with $k = 128$ to limit feature sizes to 128 dimensions. Similarly, Baseline, BCT, and LCE methods are configured with a feature dimension of 128.

| | 2 steps | | | 5 steps | | | 20 steps | | | 50 steps | | |
|---|---|---|---|---|---|---|---|---|---|---|---|---|
| METHOD | $AC$ | $AA$ | $ACA$ | $AC$ | $AA$ | $ACA$ | $AC$ | $AA$ | $ACA$ | $AC$ | $AA$ | $ACA$ |
| Baseline$_{128}$ | 0 | 24.45 | 0 | 0 | 10.43 | 0 | 0 | 3.02 | 0 | 0 | 1.51 | 0 |
| BCT$_{128}$ (Shen et al., 2020) | 1 | 31.65 | 28.83 | 0.8 | 24.08 | 16.66 | 0.01 | 14.99 | 0.11 | <0.01 | 8.32859 | 0.03 |
| LCE$_{128}$ (Meng et al., 2021) | 1 | 31.61 | 29.84 | 0.6 | 22.67 | 14.90 | 0.01 | 11.97 | 0.12 | 0 | 6.01 | 0 |
| PSP (Top-128) | 1 | 29.89 | 25.09 | 0.9 | 21.95 | 18.01 | 0.92 | 17.53 | 15.44 | 0.9 | 16.64 | 14.89 |
| LSP (Top-128) | 1 | 32.84 | 29.26 | 1 | 25.26 | 23.59 | 0.83 | 20.57 | 17.75 | 0.81 | 19.59 | 15.90 |

