# OpenReview forum: "The Probability Simplex is Compatible"
_ICLR.cc/2025/Conference — Submitted to ICLR 2025_

### Official Review · Reviewer_EQDT · 2024-10-28

**Soundness:** 3
**Presentation:** 2
**Contribution:** 2
**Rating:** 6
**Confidence:** 5

**Summary:**

The paper addresses the issue of model compatibility in representation learning, where features generated by an updated model cannot be directly compared with those of an old model. It proposes a new approach to this problem: using the (projected) softmax/logits vector as the feature vector (referred to as PSP/LSP, respectively). This approach resolves the compatibility issue. The paper also presents theoretical results showing that, under certain assumptions, using PSP/LSP as a feature ensures compatibility, as defined in the BCT paper[1]. Finally, it provides empirical evidence supporting the effectiveness of this approach on CIFAR100, ImageNet, and Google LandMarkv2 datasets.

[1] Shen, Yantao, et al. "Towards backward-compatible representation learning." Proceedings of the IEEE/CVF Conference on Computer Vision and Pattern Recognition. 2020.

**Strengths:**

- The paper tackles the model compatibility problem from a new angle: using projected softmax/logits vectors as the feature.
- Theoretical results provided in the paper appear sound under the stated assumptions.
- The introduction, related work, and problem definition in Section 3.1 are comprehensive and easy to follow.
- Experimental results demonstrate the proposed approach’s empirical efficacy, especially in cases involving multiple model updates.

**Weaknesses:**

- The paper proposes using projected softmax/logits vectors as features, resulting in compatibility out-of-the-box (i.e., without the need for additional training). However, the paper does not discuss the accuracy impact of this design choice: for a model $M$ , what would be the recall@1 when using  $M$ ’s features (for both query and gallery) versus using PSP and LSP? Could you please report and discuss test recall@1 for all models and datasets considered in the paper, including open-set setups and cases with different numbers of training classes? For example, the diagonal entries of the compatibility matrices provided in Figure 7d-f show drop in accuracy when using PSP/LSP instead of main backbone features.

- A key practical application of model compatibility in representation learning is in large-scale retrieval systems, such as face recognition. In these cases, training involves supervised learning on a large number of classes/identities ($C$  classes), while the test-time classes are unknown (i.e., an open-set setup). The proposed approach requires maintaining  $C$ -dimensional vectors, which can be very large compared to typical embedding dimension (128-256 as in [1]) and may make the proposed approach impractical. Can the authors propose/explore some dimensionality reduction mechanism to address this issue? A face recognition experimental setup (also discussed below) would be good to explore this.

- Many practical use cases involve infrequent model updates, such as updating the face recognition embedding model every few years in a large retrieval system. The proposed approach appears to perform worse than the baselines (e.g., BCT[1]) in such scenarios (2-step results in Table 1), which significantly limits its applicability for real-world problems. What would be the performance of the proposed approach for a single model update of a face recognition setup (the same as those considered in [1]) in comparison to baselines?

- Additionally, the approach does not apply to self-supervised learning, where there is no notion of logits/softmax.

- The theorem provided is based on several assumptions (listed below based on this reviewer's understanding). Could you please clearly state in the main paper the assumptions and the form of conclusion? Keeping the actual proof in the appendix is fine.
  1. Normalized projected softmax features form a vMF distribution.
  2. The variance of this vMF distribution decreases with model updates.
  3. The proof holds only in an expectation sense: on average, same-class features are closer and different-class features are farther when using the new model for queries. No probabilistic bound is provided.

- The presentation of Sections 3.2 and 3.3 is difficult to follow and could be simplified and restructured. For example, the projection matrix is used in Equation 2, but its actual definition is presented two pages later in Proposition 2.

- Additional minor issues are listed below:
  - Line 184: “Trained with stochastic gradient descent (SGD)”—clarify the significance of this statement.
  - Equation 2:  $|| \cdot ||_2$  seems to be used to indicate L2-normalization, though it typically denotes the L2-norm.
  - Line 211: Softmax is referred to as a projection, although, mathematically, the softmax function is not a projection:  $\sigma(\sigma(z)) \neq \sigma(z)$.
  - Line 304: typo.
  - Line 405: ETF abbreviation is used without definition.
  - Line 953: $\cos  \phi$  should be $\cos  \alpha$

[1] Shen, Yantao, et al. "Towards backward-compatible representation learning." Proceedings of the IEEE/CVF Conference on Computer Vision and Pattern Recognition. 2020.

**Questions:**

- In Table 1-C, some values are reported as NaN, OOM, and $\times$. Please provide additional explanation or context for these entries.
- The proposed PSP and LSP features can also be computed for certain baseline approaches that involve learning. Have the authors considered such cases? For instance, a new model could be trained with the BCT loss. In that case, would it still be beneficial to use PSP and LSP as features instead of the actual features?
- In Fig. 5(a), regarding the same-class cosine distance, is  $\alpha = 0$ ? Please clarify. If so, does the figure simply illustrate that if the variance of a vMF is reduced through an update, the average distance between vectors decreases?
- In Fig. 5(b), does it make sense to also plot  $1 - \cos{\alpha}$  for different $d$?

**Details Of Ethics Concerns:**

No ethics concern by this reviewer.

---

> ### Author Response · Authors · 2024-11-21
> **Response to the Highlighted Strenghts (EQDT)**
>
> We thank the Reviewer for the thoughtful summary they provided about our paper and for highlighting the strengths of our paper, including our novel approach to model compatibility, sound theoretical results, comprehensive coverage, and empirical efficacy as demonstrated in the experiments.

---

> ### Author Response · Authors · 2024-11-21
> **Response to the Weaknesses (EQDT) 1/2**
>
> [**Impact of compatibility and  model performance.**]
>
> We thank the Review  for the insightful comment. To further clarify, employing M's features alone for both query and gallery setups constitutes what is typically referred to as a self-test, observable in the main diagonal of the compatibility matrix.
> As correctly pointed out by the Reviewer, the comparisons suggested are already reflected within the main diagonal of the compatibility matrix presented in our study. Specifically, the open-set setups are illustrated in the Fig. 7d, 7e, 7f, as indicated. Unfortunately, to ensure compatibility, some performance trade-offs are inevitable.
>
> ---
>
> [**Exploring top-k dimensionality reduction.**]
>
> We thank the reviewer for the insightful question.
> One potential approach to mitigate the memory impact of storing features is through Top-K sparsification [1][2].
> In the following table, we present the compatibility metrics for various compression levels achieved using Top-K sparsification in a two-network setting (ResNet18 and ResNet152), pretrained on ImageNet1K and validated on the open-set CIFAR100.
> It can be observed that as the feature size decreases—corresponding to higher sparsification levels (lower values of K)—compatibility metrics such as $AA$ and $ACA$ tend to slightly decline.
>
> [1] Longfei Zheng, Yingting Liu, Xiaolong Xu, Chaochao Chen, Yuzhou Tang, Lei Wang, and Xiaolong Hu. Fedpse: Personalized sparsification with element-wise aggregation for federated learning. In Proceedings of the 32nd ACM International Conference on Information and Knowledge Management, pp. 3514–3523, 2023.
>
> [2] Longfei Zheng, Yingting Liu, Xiaolong Xu, Chaochao Chen, Yuzhou Tang, Lei Wang, and Xiaolong Hu. Fedpse: Personalized sparsification with element-wise aggregation for federated learning. In Proceedings of the 32nd ACM International Conference on Information and Knowledge Management, pp. 3514–3523, 2023.However, the AC metric remains unaffected, indicating that compatibility between the two models is preserved.
>
>
> | Method     | AC (Top-16) | AA (Top-16) | ACA (Top-16) | AC (Top-32) | AA (Top-32) | ACA (Top-32) | AC (Top-64) | AA (Top-64) | ACA (Top-64) | AC (Top-128) | AA (Top-128) | ACA (Top-128) | AC (Top-512) | AA (Top-512) | ACA (Top-512) | AC (Top-1000) | AA (Top-1000) | ACA (Top-1000) |
> |------------|-------------|-------------|--------------|-------------|-------------|--------------|-------------|-------------|--------------|--------------|--------------|---------------|--------------|--------------|---------------|---------------|---------------|----------------|
> | PSP        | 1.0         | 38.63       | 40.02        | 1.0         | 39.05       | 40.38        | 1.0         | 39.09       | 40.41        | 1.0          | 39.12        | 40.45         | 1.0          | 39.07        | 40.47         | 1.0           | 39.04        | 40.47         |
> | LSP        | 1.0         | 45.44       | 45.10        | 1.0         | 47.42       | 45.86        | 1.0         | 48.77       | 46.10        | 1.0          | 49.72        | 47.17         | 1.0          | 50.86        | 48.62         | 1.0           | 50.06        | 45.46         |
>
>
>
> ---
>
>
> [**Infrequent model updates in  retrieval systems.**]
>
> We thank the Reviewer for highlighting the performance in scenarios with infrequent updates and appreciate their focus on face recognition. In addition to face recognition with infrequent update the applicability of our approach can extend to other areas such as real-time robotics and video surveillance, where rapid updates are essential to adapt to unknown contexts. In such systems, representation updates might occur within seconds or minutes. This rapid adaptability is crucial when a robot sends real-time images to a remote learning server, requiring timely updates based on these images. Such updates ensure that the newly downloaded representation model remains compatible with the features previously extracted and stored in the robot's memory buffer.
>
> We have evaluated the performance of our approach after a single model update in a face recognition setup. This analysis also addresses Reviewer Zp2R's request for face recognition evaluation, and we invite the Reviewer to refer to that response for further details. Our approaches (particularly LSP) obtain comparable performance with BCT when the model is updated once, while outperforming it for multiple step updates.

---

> ### Author Response · Authors · 2024-11-21
> **Response to the Weaknesses (EQDT) 2/2**
>
> [**Compatibility in self-supervised learning.**]
>
> As previously noted in our response to Reviewer Zp2R, we recognize that would be very interesting to address backward compatibility, starting with models learned through self-supervised learning. However, learning compatible representations starting from two generic models, for which no assumptions are made (including those learned through self-supervised methods), typically involves making them compatible by mapping one representation onto the other.
> To learn this mapping, a specific training set is typically required. When multiple updates are made, a series of mapping transformations accumulate, which must be applied sequentially. This results in an increasingly complex architecture that reduces the computational advantage of feature extraction, as each image must traverse all the mappings. However, the BCT paper does not employ a mapping-based approach. We followed the supervised setting described in the BCT paper, which does not employ a mapping-based approach. Instead, it retrains a new model by leveraging the previous model and classifier within a supervised task in an incremental setting, using this as a surrogate from which to learn the feature representation to perform retrieval.
>
> ---
>
> [**Theorem assumptions and conclusion.**]
>
> We thank the Reviewer for the precise observations regarding the assumptions underlying the theorem presented in our manuscript.
> We agree with your understanding, the revised paper now includes clear statements of the assumptions in the main body and outlines the form of the conclusion succinctly. The detailed proof remains in the appendix, as previously structured.
>
> ---
>
> [**Improved presentation.**]
>
> We thank the Reviewer for the feedback regarding Sections 3.2 and 3.3. We have revised these sections for clarity and restructured the content, ensuring the projection matrix is defined before its use in Equation 2. These changes should improve coherence and understanding. We appreciate the suggestions.
>
>
> ---
>
> [**Fixed minor issues.**]
>
> After consideration, we agree that specifying SGD is not crucial in the context of our discussion, as it refers to standard training procedures in deep neural networks. We have omitted this detail to streamline the text. Thank you for pointing this out.
>
> We thank the Reviewer for pointing out the confusion. It was indeed intended to denote L2-norm, not L2-normalization. We apologize for this oversight and for any confusion it may have caused. The text has been corrected to clearly reflect its intended meaning. We thank the Reviewer for their precise feedback.
>
> We thank the Reviewer for highlighting the incorrect terminology used to describe the softmax function on Line 211. We acknowledge that referring to softmax as a projection was inaccurate, as it does not satisfy the mathematical property $\sigma(\sigma(z)) \neq \sigma(z)$. The misunderstanding arose because the softmax outputs sum to 1, positioning them on a hyperplane, which might suggest a projection-like behavior. We have corrected this in the manuscript to ensure precision in our descriptions. We are grateful for the collaboration and appreciate your careful reading and valuable correction.

---

> ### Author Response · Authors · 2024-11-21
> **Response to the Questions (EQDT) 1/2**
>
> [**Additional Explanation about NaN, OOM and x symbols.**]
>
> We thank the Reviewer for requesting clarification.
> The ''NaN'' issue arises from the instability associated with adversarial training, a challenge widely recognized in the literature [1][2][3]. This condition may necessitate hyperparameter tuning or the use of a pretrained network, e.g. similar to the training setting used in the Google Landmark v2 dataset.
>
> Regarding the ''OOM'' value, it denotes a GPU out-of-memory error. This occurs due to the need to instantiate a simplex matrix of size $O(N^2)$, where $N$ is the number of preallocated classes. Regrettably, our current hardware was inadequate to support this demand.
>
> Lastly, the ''$\times$'' designation indicates that the ETF cannot be evaluated on the Google Landmark v2 dataset because the number of classes (81313) exceeds the encoder’s output limit of 2048.
>
> [1] Ian Goodfellow, Jean Pouget-Abadie, Mehdi Mirza, Bing Xu, David Warde-Farley, Sherjil Ozair, Aaron Courville, and Yoshua Bengio. Generative adversarial nets. Advances in neural information processing systems, 27, 2014.
>
> [2] Martin Arjovsky and L´eon Bottou. Towards principled methods for training generative adversarial networks. arXiv preprint arXiv:1701.04862, 2017.
>
> [3] William Fedus, Mihaela Rosca, Balaji Lakshminarayanan, Andrew M Dai, Shakir Mohamed, and Ian Goodfellow. Many paths to equilibrium: Gans do not need to decrease a divergence at every step. arXiv preprint arXiv:1710.08446, 2017.
>
> ---
>
> [**Using PSP/LSP features in BCT.**]
>
> We thank the Reviewer for suggesting the consideration of PSP and LSP features for baseline approaches involving learning. This suggestion is very interesting and appreciated. After evaluation, we observed that incorporating these features in a scenario with one model update results in a slight improvement.
> However, there remains an issue in tasks involving five steps where the regularization by BCT does not provide a global configuration of the feature space which remains aligned during training, thus compromising compatibility.
> Nevertheless, using features derived from softmax outputs and logits of models trained with BCT (BCT with PSP and BCT with LSP, respectively) achieves better performance compared to standard BCT in terms of higher compatibility ($AC$) and comparable accuracy ($AA$) values.
> We appreciate this valuable input, which has enriched our analysis. This also demonstrates that our approach can be used in conjunction with others, improving their compatibility at the cost of a slight drop in accuracy performance.
>
> | Method      | 2 tasks AC | 2 tasks AA | 2 tasks ACA | 5 tasks AC | 5 tasks AA | 5 tasks ACA |
> |-------------|------------|------------|-------------|------------|------------|-------------|
> | BCT         | 1          | 46.40      | 50.21       | 0.40       | 29.93      | 11.78       |
> | PSP         | 1          | 36.31      | 29.05       | 0.90       | 26.04      | 21.76       |
> | LSP         | 1          | 41.14      | 36.38       | 0.70       | 30.36      | 21.91       |
> | BCT w PSP   | 1          | 38.00      | 31.28       | 0.60       | 24.63      | 17.41       |
> | BCT w LSP   | 1          | 43.18      | 43.62       | 0.60       | 28.53      | 17.80       |
>
> ---
>
> [**Decrease of average distance between vectors**]
>
> We thank the Reviewer for their insights. The Reviewer is correct; in Fig. 5(a), $\alpha = 0$, indicating that reducing the variance of a vMF distribution decreases the average angle between vectors sampled from one distribution and those from another.

---

> ### Author Response · Authors · 2024-11-21
> **Response to the Questions (EQDT) 2/2**
>
> [**Decrease of average distance between vectors. Plot.**]
>
> Considering $\cos \alpha$  alone (without the concentration parameters) is equivalent to assuming that deterministic class features follow a von Mises-Fisher (vMF) distribution with infinite concentration around their means. In this ideal scenario, all the images belonging to a specific class can be represented exactly by the same feature, and introducing additional data cannot improve this representation since the features representation is already perfect. However, as defined in BCT, compatibility aims to determine if more data—which leads to less uncertain features—can improve both intra-class and inter-class distances compared to the previous model. Nonetheless, if the features are already perfect, no additional data can further reduce their uncertainty. According to this, the two concentration parameters of the vMF distributions are essential and cannot be omitted, as they fundamentally quantify the relative expressive power between a base and an updated model. In this sense $1 - \cos \alpha$ for different $d$ values can represent the theoretical limit as the number of updates approaches infinity, where infinite data leads to perfect representation.
> Including this in the plots could clarify how each configuration progresses towards optimal model performance. However, further refinement of this theoretical aspect seems to be required and is being actively explored as part of the ongoing research.

---

> ### Comment · Reviewer_EQDT · 2024-11-23
> **Post rebuttal discussion**
>
> The reviewer thanks the authors for their detailed and comprehensive response, addressing the points of confusion, providing additional experimental results, and improving the quality of the paper. A few points are further discussed below:
>
> **Impact of compatibility and model performance.**
> Using projected softmax/logits as features (in the self-test setup) results in accuracy degradation. The reviewer acknowledges that a similar concept exists for regularization-based methods such as BCT, where the performance of the new model learned with regularization is worse compared to a freely trained new model. However, this is not the case for all compatibility approaches, such as those using a mapping method like FCT[1]. Additionally, while using projected softmax/logits allows training-free compatibility out of the box, it does not support (partial) backfilling (i.e., if one wants to pay additional cost to gain more accuracy, that would not be possible), as discussed in [2].
>
> **Exploring top-k Sparsification dimensionality reduction.**
> Thank you for providing results using Top-k Sparsification as a dimensionality reduction approach. Including these results, along with the additional points mentioned below, should improve the paper’s contribution:
>  - Elaborate Top-K Sparsification and how it fits into the compatibility setup.
>  - When possible, compare with baselines using regular features of the same dimension (e.g., for K=256, how does it compare with BCT with 256-dimensional features?).
>
> **Summary.**
> This reviewer can summarize the specific contributions as follows and encourages the authors to clarify the pros/cons/limitations in the paper:
> - Positives:
>   - A new approach to model compatibility that does not require any training.
>   - Good empirical results when the model is updated very frequently.
>   - Possibility of dimensionality reduction (e.g., using Top-k Sparsification).
> - Limitations:
>   - Accuracy decline due to using projected softmax/logits instead of regular features.
>   - Limited benefits to scenarios with very frequent updates only.
>   - Need for dimensionality reduction when the number of classes is large.
>   - Infeasibility of backfilling/partial-backfilling when accuracy is prioritized over cost.
>
> Based on the above, this reviewer would be happy to increase the score to 6 after the rebuttal.
>
> [1] Ramanujan, Vivek et al. "Forward compatible training for representation learning." Proceedings of the IEEE/CVF Conference
> on Computer Vision and Pattern Recognition, 2022.
>
> [2] Jaeckle, Florian, et al. "Fastfill: Efficient compatible model update." The Eleventh International Conference on Learning Representations (2023).

---

> ### Author Response · Authors · 2024-11-27
> **Response to Post Rebuttal discussion (EQDT) 1/2**
>
> We thank the Reviewer for the constructive feedback and for acknowledging our efforts to clarify the points of confusion and improve the paper's quality. We appreciate the Reviewers' positive adjustment of the score.
>
> ---
>
> [**Discussion about the Partial Backfilling**]
>
>
> We agree with the Reviewer that mapping-based methods may have a lesser impact on degradation compared to BCT-like approaches. We also acknowledge that, in its current formulation, our proposed method cannot support online-backfilling like Forward Compatible Training, since it does not assume the availability of additional data to learn a forward mapping. However, from a practical perspective, it should be feasible to incorporate a learnable residual mapping from a side-training dataset into our non-trainable approach.
> This could improve compatibility by introducing minor corrections to our proposed framework.
> If these corrections are minor, simpler and less expressive forward mapping transformations could be utilized, potentially requiring less data. Further refinement and extensive evaluation of this improvement appear necessary and are currently under active investigation as part of our ongoing research efforts.
>
> Regarding backfilling, our method enables partial backfilling by transforming some features extracted from the base model using our approach, while re-extracting the remaining features with an updated model that is trained independently, potentially elsewhere. This process still achieves time savings in reindexing.
> According to this, the following table presents the Recall@1 values for LSP and PSP features across different backfill ratios between a gallery extracted with ResNet18 and a query set extracted with ResNet152.
> The gallery is partially backfilled with representations from ResNet152.
> This backfilling can be performed asynchronously with the operation of the retrieval system. We employ a naive approach (random selection) to backfill the gallery set. Upon completion of the backfill, the proposed approach achieves the performance of the independently trained model.
>
> | Backfill Ratio (\%) | 0      | 10     | 20     | 30     | 40     | 50     | 60     | 70     | 80     | 90     | 100    |
> |---------------------|--------|--------|--------|--------|--------|--------|--------|--------|--------|--------|--------|
> | PSP                 | 0.4049 | 0.4261 | 0.4349 | 0.4452 | 0.4508 | 0.4605 | 0.4659 | 0.4667 | 0.4685 | 0.4725 | 0.4757 |
> | LSP                 | 0.4546 | 0.5035 | 0.5333 | 0.5632 | 0.5817 | 0.5868 | 0.5903 | 0.6022 | 0.606  | 0.6075 | 0.6148 |

---

> ### Author Response · Authors · 2024-11-27
> **Response to Post rebuttal discussion (EQDT) 2/2**
>
> [**Elaboration of Top-k Sparsification**]
>
>
> In response to the Reviewer’s request for further elaboration on top-$k$ sparsification for dimensionality reduction, it is worth noting that this approach is particularly justified for PSP feature representations. Indeed, top-$k$ selectively ignores components of the softmax output vectors that are minimally activated. The retained $k$ components correspond to classes most similar to the input class. Thus, in an open-set scenario, these components might represent an unseen class, indicated by the top-$k$ classes that show some activation in the softmax output.
>
> From a theoretical perspective, [1] demonstrates that softmax loss yields competitive top-$k$ performance for *all* values of $k$ simultaneously. This indicates that it preserves the ranking of the probabilities, which can contribute to keep the order of the activated components across different dimensions when they are transformed by our PSP method.
> Accordingly, top-$k$ sparsification seems to be well-suited for dimensionality reduction by setting all entries of a PSP feature representation vector to zero except for the top-$k$. Although LSP lacks similar theoretical support because logits are not probabilistically normalized, our alignment proof with respect to PSP, and the fact that both share similar hyperspherical geometry, suggests that they might behave similarly.
>
> In the following table, we report results for PSP and LSP, where top-$k$ dimensional reduction has been applied to reduce features to the top-128 dimensions. For BCT and LCE, the feature size is reduced to the same 128 dimensions via a fully-connected layer, facilitating a direct comparison of the two methods under the same feature dimensions. These models were trained on the TinyImageNet200 training set and evaluated on the TinyImageNet200 validation set using Recall@1 at 2, 5, 20, and 50 update steps. Both PSP and LSP show slightly improved compatibility (AC) and accuracy (AA) performance in all cases compared to the results in Tab. 1b of the revised paper, where no dimension reduction was applied.
> Conversely, all compared methods exhibit a decrease in compatibility and accuracy due to the constrained feature space. Specifically, for BCT and LCE, performance is comparable to that in Tab. 1b of the revised paper during infrequent updates (2 and 5 steps) but declines significantly with more frequent updates.
>
> | Method              | AC 2 steps | AA 2 steps | ACA 2 steps | AC 5 steps | AA 5 steps | ACA 5 steps | AC 20 steps | AA 20 steps | ACA 20 steps | AC 50 steps | AA 50 steps | ACA 50 steps |
> |---------------------|------------|------------|-------------|------------|------------|-------------|-------------|-------------|--------------|-------------|-------------|--------------|
> | Baseline_128        | 0          | 24.45      | 0           | 0          | 10.43      | 0           | 0           | 3.02        | 0            | 0           | 1.51        | 0            |
> | BCT_128 [Shen2020]  | 1          | 31.65      | 28.83       | 0.8        | 24.08      | 16.66       | 0.01        | 14.99       | 0.11         | <0.01       | 8.32        | 0.03         |
> | LCE_128 [Meng2021]  | 1          | 31.61      | 29.84       | 0.6        | 22.67      | 14.90       | 0.01        | 11.97       | 0.12         | 0           | 6.01        | 0            |
> | PSP (Top-128)       | 1          | 29.89      | 25.09       | 0.9        | 21.95      | 18.01       | 0.92        | 17.53       | 15.44        | 0.92        | 12.82       | 11.89        |
> | LSP (Top-128)       | 1          | 32.84      | 29.26       | 1          | 25.26      | 23.59       | 0.83        | 20.57       | 17.75        | 0.81        | 19.59       | 15.90        |
>
>
> Although the representation benefits from top-$k$ sparsification for dimensionality reduction, as empirically confirmed (Appendix K), this promising and important aspect merits further investigation in future studies.
>
> [1] Lapin, M., Hein, M., and Schiele, B. "Loss functions for top-$k$ error: Analysis and insights". (CVPR2016)

---

### Official Review · Reviewer_Zp2R · 2024-10-29

**Soundness:** 2
**Presentation:** 2
**Contribution:** 2
**Rating:** 3
**Confidence:** 4

**Summary:**

This paper proposes a solution to deal with the backwards compatibility problems -- when better models, new tasks, etc., necessitates updating the model in production, it takes a lot of effort to ensure the new model is compatible with the old model in production. This paper proposes a solution that project softmax and logits into the simplex space, and "reconciling" the new and old model there.

**Strengths:**

- The idea proposed in this paper is interesting. Without even involving the old model in the new model training, which most backward compatible solutions need, the proposed solution projects either the new classes/tasks into the probability simplex, thereby achieving backward compatibility
- The paper also presents two types of projections, namely the softmax/logits projections. Analysis of the behavior and performance of both types of projections are given.
- This paper also highlights an interesting aspect of backward compatibility, which is whether we can do it without needing the old model. That means that we can make any two models backward compatible to each other without them knowing of each other when they were trained and built.

**Weaknesses:**

- The paper can be written better. The paper talks about a P_{t,k}, P_{k,k} projection functions that take the softmax and project into the simplex space. However, I am not able to find anywhere how these projections came about, how they are learned/trained. If the authors can point me to where in the paper describes how these projections are learned, I would appreciate it. I did comb through the appendix quickly as well as the supp, but did not seem to find it. Otherwise, I may have been confused by your writing.
- The paper only consider the task incremental setting in the formulation, making it a bit hard for me to understand how this is extended to retrieval task or when model architecture is updated (Table 2). For example, what if we are dealing with a representation self-supervised model and the retrieval task, which most BCT papers are dealing with, and a newer model comes along?
- I find the evaluation metrics a bit obscure, namely the average compatibility, average accuracy and average compatibility accuracy. Is it possible for the authors to provide numbers before and after you add classes or change architecture per method, so that I can tell whether the old performance is maintained. Specifically, before you add classes, the performance of the new model should maintain on the old classes.
- Is it possible to add retrieval task and benchmark with baselines? A lot of BCT papers use retrieval to measure their performance. It is hard for me to tell without any retrieval task (I am assuming Table 1 and 2 are all classification results, iiuc).
- Table 2 needs to be compared to other benchmarks.
- I would also like to see a lot more modern architecture benchmarking besides just a Max ViT_T in Table 2. For example, a transition of architectures from ViT-S, to ViT-B, huge, etc.
- As it stands, it is hard to know whether the paper's method works since Table 1 and 2 are not 100% showing that the method always beat the others.

**Questions:**

See weaknesses.

---

> ### Author Response · Authors · 2024-11-21
> **Response To the Highlighted Strenghts (Zp2R)**
>
> We thank the Reviewer for recognizing the strengths of our work: specifically the novel approach to backward compatibility without involving the old model, the direct analysis of softmax and logits projections, and the potential of achieving compatibility between models trained independently.
> Additionally, we would like to emphasize that our proposed approach has theoretical support, further validating its effectiveness.

---

> ### Author Response · Authors · 2024-11-21
> **Response To Weaknesses (Zp2R) (1/2)**
>
> **[Clarification about projection matrices and their formulation.]**
>
> We thank the Reviewer for their valuable feedback and for pointing out the need for clearer explanations regarding the projection matrices $ \mathbf{P} _ {t,k} $ and $ \mathbf{P} _ {k,k} $ used in our paper. We appreciate your careful reading and acknowledge the need for clearer exposition on this aspect.
>
> To clarify, the projection matrices are \textit{not} learned or trained but are derived analytically as part of our methodology. Specifically, the matrix $ \mathbf{P} _ {k,k} $ can be explained through the concept of the centering matrix [marden96]. According to this, our proposed projection matrix can be expressed as:
>
> $$
> \mathbf{P} _ {k,k} = \mathbf{I} _ {C^k} - \frac{1}{C^k} \mathbf{J} _ {C^k}
> $$
>
> where $\mathbf{I} _ {C^k}$ is the identity matrix, $ C^k $ represents the number of elements, and $ \mathbf{J}_{k,k} $ is a matrix entirely composed of ones. The projection $ \mathbf{P} _ {t,k} $ can be then defined as:
>
> $$
> \mathbf{P} _ {t,k} = \left[ \mathbf{P} _ {k,k} \mid  \mathbf{0} \right]
> $$
>
> where where $\mathbf{0} \in \mathbb{R}^{(C^t-C^k) \times C^k}$  with $C^t$ and $C^k$ the number of classes of the updated and base model, respectively. We realize this explanation was not adequately highlighted in the paper, and we have revised the relevant sections to ensure this is clearly presented and easy to locate. We believe that this adjustment will address the raised concerns and clarify the theoretical foundation of the projection matrices.
>
> **[Marden96]** [John I Marden. Analyzing and modeling rank data. CRC Press, 1996.]
>
> ---
>
> **[Clarification about training setting, retrieval and self-supervision]**
>
> We thank the Reviewer for the insightful comments regarding the incremental setting discussed in our paper. We followed the standard retrieval-based incremental setting pioneered by BCT [1], which has been widely adopted in subsequent research. To provide further clarification, we emphasize that our approach and its associated theoretical analysis have not deviated from the original formulation presented in [1]. There, classification was used as a surrogate task to learn representations, which were then evaluated in a retrieval setting. In addition, one of our main contributions is discovering how to achieve compatibility in independently trained models (as in Tab. 2), a novel approach not addressed by BCT. This notably differentiates our work from the task incremental setting of BCT and related papers.
>
> We recognize that it would be very interesting to address backward compatibility, starting with models learned through self-supervised learning. However, learning compatible representations starting from two generic models, for which no assumptions are made (including those learned through self-supervised methods), typically involves making them compatible by mapping one representation onto the other (as for example in Meng, Qiang, et al. "Learning compatible embeddings." ICCV2021).
>
> To learn this mapping, a specific training set is typically required. When multiple updates are made, a series of mapping transformations accumulate, which must be applied sequentially. This results in an increasingly complex architecture that reduces the computational advantage of feature extraction, as each image must traverse all the mappings. However, the BCT paper does not employ a mapping-based approach; instead, it retrains a new model by leveraging the previous model and classifier within a supervised task in an incremental setting, using this as a surrogate from which to learn the feature representation to perform retrieval.
>
> ---
>
> **[Clarification about compatible metrics and performance.]**
> We thank the Reviewer for the comments on the evaluation metrics used in our manuscript. We appreciate the Reviewer's perspective that metrics such as average compatibility, average accuracy, and average compatibility accuracy might seem complex. We acknowledge this complexity but also want to point out that the detailed results for these metrics, including the impact on performance when new classes are added or the architecture is changed, are indeed inherently included in those metrics and therefore present in our reported data. Specifically, the main diagonals of the compatibility matrices align with the Reviewer's intuition of ``before and after'', as they quantify performance gains from using a learned model to perform retrieval.
> When both the query and gallery are extracted using the same model, these diagonals directly illustrate how the classification surrogate task contributes to the learned representation's performance.
> In Appendix H of the revised manuscript, we have included the compatibility matrices for PSP, LSP, and all compared methods to further highlight this, specifically for the 5-step scenario on the CIFAR100 dataset under the extended classes setting.
>
> ---

---

> ### Author Response · Authors · 2024-11-21
> **Response To Weaknesses (Zp2R) (2/2)**
>
> **[Evaluation metrics before and after.]**
>
> We thank the Reviewer for the feedback on the evaluation metrics. We would like to clarify that Tables 1 and 2 present retrieval results, not classification outcomes. We acknowledge the need for clearer communication on this aspect, which is more explicitly stated in our revised manuscript.
>
> ---
>
> **[ Compatibility with ViT architectures.]**
>
> We appreciate the Reviewer's suggestion to compare our results with a broader range of modern architectures. In response to the Reviewer's comment, we have included additional benchmarks in our revised manuscript, encompassing a variety of architectures including the Vision Transformer (ViT).
> In Appendix J of the revised manuscript, we provide detailed analyses demonstrating that PSP and LSP consistently achieve the highest compatibility scores across various datasets, including ImageNet1k, Cifar100, and Places365.
>
> ---
>
> **[ Advantages and accuracy trade-off.]**
>
> We thank the reviewer for their feedback on the effectiveness of our method, as demonstrated in Tables 1 and 2. While we acknowledge their observations, it is important to highlight that our method offers significant advantages: it requires no retraining, no prior BCT classifier, no auxiliary loss as in BCT, and neither mappings nor any modifications to the network architecture. Moreover, unlike traditional BCT approaches that rely on transitive properties and may not sustain performance over the long term, our approach maintains its effectiveness, albeit with an increased representation dimension. Thanks to the suggestion of colleague Reviewer EQDT, we have shown that top-k sparse representation can reduce the dimension without significantly compromising performance.
>
> We also wish to highlight that our main contribution includes providing theoretical support for the proposals discussed. We believe this enhances the foundation of our findings.

---

> ### Comment · Reviewer_Zp2R · 2024-11-25
>
> Thanks for your clarifications.
>
> I already understood and appreciate that this method does not require re-training like BCT when I first read the paper.
>
> However, my biggest concern remains that you have designed metrics that are different from what was used in baselines i.e. before/after (please look at the BCT paper for example). This makes it hard for me to judge the validity of this proposed approach. IIUC, you should be able to extract the same metrics that ALL the baselines used based on your metrics and present to the reviewers here. In fact, if you were to check out say BT2 paper, they have very comprehensive before and after results that present an intuitive report of how well it works e.g., Table 6 in BT2. I wish to see the same thing.

---

> > ### Author Response · Authors · 2024-11-27
> > **Response to the Official Comment by Reviewer (Zp2R) 1/2**
> >
> > We thank the reviewer for acknowledging and understanding the distinct aspects of our method as initially presented in the paper.
> >
> > We appreciate the Reviewer's concern about the metrics used, which differ from those in the BCT and BT2 papers. We acknowledge that these differences may raise concerns. Our choice of metrics was driven by the need to provide a comprehensive summary of performance across frequent updates, encapsulating the overall effectiveness of each method in a single metric throughout all updates. In scenarios involving numerous updates, using a single table can be problematic not only because of the required space but also because it does not adequately reflect the overall performance observed across all updates.
> >
> > For example, the paper [1] shows a $3 \times 3$ compatibility matrix in the Supplementary Material to illustrate multiple sequential updates. However, it includes only two update steps, and assessing the relative performance improvement at each step relies on visual inspection, which can be challenging to decode and summarize. To effectively evaluate multiple updates without extensive visual analysis of large compatibility matrices, we adopted the metrics AC, AA and ACA from the recent paper [2]. These metrics "summarize" the performance in the compatibility matrix by extracting a single number that represents overall performance, aiming for a balance between being overly compatible (i.e., avoiding updates entirely) and sacrificing the expressiveness of the current model by excessively mimicking the outdated, lower-performing model. This evaluation approach resembles the Class-Incremental Learning metrics, as originally introduced in [3], where multiple updates are performed and overall performance is typically summarized into a single number representing the change in classification performance before and after each task update. In the revised manuscript, we briefly clarify this concept prior to defining the metrics.
> >
> > In response to the Reviewer's request, we present the performance data in compatibility matrices of Figs. 8a, 8b, and 8c in the revised paper, formatted in the style of Table 6 from BT2, to ensure consistency and comparability.
> > This format directly addresses the desire to see comparable data.
> > To ensure consistency with the notation used in the BT2 paper, the subscripts have been revised to indicate the network architectures employed for model updates, instead of the time-steps previously specified.
> > These architectures include: AlexNet ($\phi _ {alex}$) at step 1, ResNet50 ($\phi _ {res50}$) at step 2, RegNetX\_3.2GF ($\phi _ {regnet}$) at step 3, ResNet152 ($\phi _ {res152}$) at step 4, and MaxViT\_T ($\phi _ {maxvit}$) at step 5.
> > This format not only clarifies the presentation of architecture names and their performance values but also directly illustrates how performance evolves before and after model updates across various architectures.

---

> > ### Author Response · Authors · 2024-11-27
> > **Response to the Official Comment by Reviewer (Zp2R) 2/2**
> >
> > | Comparison Pair    | Recall@1 |
> > |--------------------|----------|
> > | BASELINE           |          |
> > | $\phi _ {alex} / \phi _ {alex}$  | 27.26    |
> > | $\phi _ {res50} / \phi _ {alex}$  | 0.06     |
> > | $\phi _ {regnet} / \phi _ {alex}$  | 0.08     |
> > | $\phi _ {res152} / \phi _ {alex}$  | 0.12     |
> > | $\phi _ {maxvit} /  \phi _ {alex}$ | 0.09     |
> > | $\phi _ {res50} / \phi _ {res50}$  | 66.33    |
> > | $\phi _ {regnet} / \phi _ {res50}$  | 0.10     |
> > | $\phi _ {res152} / \phi _ {res50}$  | 0.04     |
> > | $\phi _ {maxvit} /  \phi _ {res50}$ | 0.11     |
> > | $\phi _ {regnet} / \phi _ {regnet}$  | 73.68    |
> > | $\phi _ {res152} /  \phi _ {regnet}$ | 0.17     |
> > | $\phi _ {maxvit} /  \phi _ {regnet}$ | 0.10     |
> > | $\phi _ {res152} / \phi _ {res152}$  | 75.38    |
> > | $\phi _ {maxvit} /  \phi _ {res152}$ | 0.10     |
> > | $\phi _ {maxvit} /  \phi _ {maxvit}$ | 80.86    |
> > | PSP                |          |
> > | $\phi _ {alex} / \phi _ {alex}$  | 49.22    |
> > | $\phi _ {res50} / \phi _ {alex}$  | 69.22    |
> > | $\phi _ {regnet} / \phi _ {alex}$  | 73.15    |
> > | $\phi _ {res152} / \phi _ {alex}$  | 49.39    |
> > | $\phi _ {maxvit} /  \phi _ {alex}$ | 75.51    |
> > | $\phi _ {res50} / \phi _ {res50}$  | 76.65    |
> > | $\phi _ {regnet} / \phi _ {res50}$  | 70.55    |
> > | $\phi _ {regnet} / \phi _ {res50}$  | 74.62    |
> > | $\phi _ {res152} / \phi _ {res50}$  | 76.53    |
> > | $\phi _ {maxvit} /  \phi _ {res50}$ | 77.83    |
> > | $\phi _ {regnet} / \phi _ {regnet}$  | 76.51    |
> > | $\phi _ {res152} /  \phi _ {regnet}$ | 78.56    |
> > | $\phi _ {maxvit} /  \phi _ {regnet}$ | 79.73    |
> > | $\phi _ {res152} / \phi _ {res152}$  | 78.31    |
> > | $\phi _ {maxvit} /  \phi _ {res152}$ | 79.42    |
> > | $\phi _ {maxvit} /  \phi _ {maxvit}$ | 79.35    |
> > | LSP                |          |
> > | $\phi _ {alex} / \phi _ {alex}$  | 32.71    |
> > | $\phi _ {res50} / \phi _ {alex}$  | 40.83    |
> > | $\phi _ {regnet} / \phi _ {alex}$  | 43.16    |
> > | $\phi _ {res152} / \phi _ {alex}$  | 49.39    |
> > | $\phi _ {maxvit} /  \phi _ {alex}$ | 39.05    |
> > | $\phi _ {res50} / \phi _ {res50}$  | 63.67    |
> > | $\phi _ {regnet} / \phi _ {res50}$  | 63.82    |
> > | $\phi _ {res152} / \phi _ {res50}$  | 66.57    |
> > | $\phi _ {maxvit} /  \phi _ {res50}$ | 62.95    |
> > | $\phi _ {regnet} / \phi _ {regnet}$  | 73.21    |
> > | $\phi _ {res152} /  \phi _ {regnet}$ | 72.63    |
> > | $\phi _ {maxvit} /  \phi _ {regnet}$ | 73.31    |
> > | $\phi _ {res152} / \phi _ {res152}$  | 75.30    |
> > | $\phi _ {maxvit} /  \phi _ {res152}$ | 74.41    |
> > | $\phi _ {maxvit} /  \phi _ {maxvit}$ | 79.24    |
> >
> >
> > [1] Meng, Q., Zhang, C., Xu, X., and Zhou, F. "Learning compatible embeddings". (ICCV2021).
> >
> > [2] Biondi, N., Pernici, F., Ricci, S., and Del Bimbo, A. "Stationary Representations: Optimally Approximating Compatibility and Implications for Improved Model Replacements". (CVPR2024).
> >
> > [3] Lopez-Paz, David, and Marc'Aurelio Ranzato. "Gradient episodic memory for continual learning." (NeurIPS2017).

---

> ### Comment · Reviewer_Zp2R · 2024-11-28
>
> Thanks for the response.
>
> However, I am not sure we are on the same page. I am asking for the same metric used by prior work, BCT, BT2, etc, to be used to compare PSP/LSP against them, i.e., a table where you have PSP, LSP, BCT, Bt2, etc, on different rows and with the before/after just like how the prior work evaluate their work. So what is the baseline here? Is the one mentioned on L416? If so, that's not what I am looking for.
>
> I will keep my score as a result.

---

> > ### Author Response · Authors · 2024-11-29
> > **Response to the Official Comment by Reviewer (Zp2R) 2/3**
> >
> > **CIFAR100:**
> >
> > | **Method**          | **Case**                                      | **CMC (top-1 - top-5)** | **Compatible?** |
> > |---------------------|-----------------------------------------------|-------------------------|-----------------|
> > | **Baseline**        | $\phi _ {\rm old}/\phi _ {\rm old}$               | 0.35 - 0.59             |                 |
> > |                     | $\phi _ {\rm new}/\phi _ {\rm old}$               | 0.01 - 0.07             | No              |
> > |                     | $\phi _ {\rm new}/\phi _ {\rm new}$               | 0.52 - 0.73             |                 |
> > | **BCT$_{128}$**     | $\phi _ {\rm old}^{\rm BCT}/\phi _ {\rm old}^{\rm BCT}$ | 0.35 - 0.59       |                 |
> > |                     | $\phi _ {\rm new}^{\rm BCT}/\phi _ {\rm old}^{\rm BCT}$ | 0.46 - 0.68       | Yes             |
> > |                     | $\phi _ {\rm new}^{\rm BCT}/\phi _ {\rm new}^{\rm BCT}$ | 0.54 - 0.71       |                 |
> > | **CoReS**           | $\phi _ {\rm old}^{\rm CoReS}/\phi _ {\rm old}^{\rm CoReS}$ | 0.33 - 0.59    |                 |
> > |                     | $\phi _ {\rm new}^{\rm CoReS}/\phi _ {\rm old}^{\rm CoReS}$ | 0.33 - 0.58    | No              |
> > |                     | $\phi _ {\rm new}^{\rm CoReS}/\phi _ {\rm new}^{\rm CoReS}$ | 0.49 - 0.72    |                 |
> > | **ETF-CE**          | $\phi _ {\rm old}^{\rm ETF-CE}/\phi _ {\rm old}^{\rm ETF-CE}$ | 0.34 - 0.58   |                 |
> > |                     | $\phi _ {\rm new}^{\rm ETF-CE}/\phi _ {\rm old}^{\rm ETF-CE}$ | 0.28 - 0.50   | No              |
> > |                     | $\phi _ {\rm new}^{\rm ETF-CE}/\phi _ {\rm new}^{\rm ETF-CE}$ | 0.52 - 0.73   |                 |
> > | **ETF-DR**          | $\phi _ {\rm old}^{\rm ETF-DR}/\phi _ {\rm old}^{\rm ETF-DR}$ | 0.30 - 0.51   |                 |
> > |                     | $\phi _ {\rm new}^{\rm ETF-DR}/\phi _ {\rm old}^{\rm ETF-DR}$ | 0.24 - 0.39   | No              |
> > |                     | $\phi _ {\rm new}^{\rm ETF-DR}/\phi _ {\rm new}^{\rm ETF-DR}$ | 0.53 - 0.67   |                 |
> > | **LCE$_{128}$**     | $\phi _ {\rm old}^{\rm LCE}/\phi _ {\rm old}^{\rm LCE}$ | 0.35 - 0.59      |                 |
> > |                     | $\phi _ {\rm new}^{\rm LCE}/\phi _ {\rm old}^{\rm LCE}$ | 0.39 - 0.65      | Yes             |
> > |                     | $\phi _ {\rm new}^{\rm LCE}/\phi _ {\rm new}^{\rm LCE}$ | 0.51 - 0.72      |                 |
> > | **AdvBCT$_{128}$**  | $\phi _ {\rm old}^{\rm AdvBCT}/\phi _ {\rm old}^{\rm AdvBCT}$ | 0.34 - 0.58  |                 |
> > |                     | $\phi _ {\rm new}^{\rm AdvBCT}/\phi _ {\rm old}^{\rm AdvBCT}$ | 0.32 - 0.60  | No              |
> > |                     | $\phi _ {\rm new}^{\rm AdvBCT}/\phi _ {\rm new}^{\rm AdvBCT}$ | 0.39 - 0.64  |                 |
> > | **BT$^2$$_{128}$**  | $\phi _ {\rm old}^{\rm BT^2}/\phi _ {\rm old}^{\rm BT^2}$ | 0.35 - 0.59     |                 |
> > |                     | $\phi _ {\rm new}^{\rm BT^2}/\phi _ {\rm old}^{\rm BT^2}$ | 0.38 - 0.61     | Yes             |
> > |                     | $\phi _ {\rm new}^{\rm BT^2}/\phi _ {\rm new}^{\rm BT^2}$ | 0.51 - 0.67     |                 |
> > | **PSP (Ours)**      | $\phi _ {\rm old}^{\rm PSP}/\phi _ {\rm old}^{\rm PSP}$ | 0.27 - 0.51      |                 |
> > |                     | $\phi _ {\rm new}^{\rm PSP}/\phi _ {\rm old}^{\rm PSP}$ | 0.29 - 0.55      | Yes             |
> > |                     | $\phi _ {\rm new}^{\rm PSP}/\phi _ {\rm new}^{\rm PSP}$ | 0.53 - 0.75      |                 |
> > | **LSP (Ours)**      | $\phi _ {\rm old}^{\rm LSP}/\phi _ {\rm old}^{\rm LSP}$ | 0.32 - 0.57      |                 |
> > |                     | $\phi _ {\rm new}^{\rm LSP}/\phi _ {\rm old}^{\rm LSP}$ | 0.36 - 0.61      | Yes             |
> > |                     | $\phi _ {\rm new}^{\rm LSP}/\phi _ {\rm new}^{\rm LSP}$ | 0.55 - 0.79      |                 |

---

> ### Author Response · Authors · 2024-11-29
> **Response to the Official Comment by Reviewer (Zp2R) 1/3**
>
> We thank the Reviewer for their clarification and appreciate their patience. We apologize for any confusion. It appears we misunderstood the Reviewer's request for a comparative analysis using the same metrics as those employed in prior works. Incorrectly, we suggested examining the main diagonals of the compatibility matrices of several methods for this comparison.
>
> We now understand that the Reviewer is requesting the standard metrics as reported by the BCT and the BT2 papers, typically presented as $\phi _ {new}/\phi _ {old}$ and $\phi _ {new}/\phi _ {new}$ by BT2, and as $(\phi _ {new}, \phi _ {old})$ and $(\phi _ {new}, \phi _ {new})$ by BCT, respectively referred to as *cross-test* and *self-test*.
> We want to assure the Reviewer that these numbers are already included in our paper, albeit in a different format. Our format choice reflects the paper's focus on frequent updates, which involve multiple old models. Consequently, instead of using subscripts $new$ and $old$, we use indices corresponding to the time steps.
>
> In the BT2 notation, $\phi _ {new}/\phi _ {new}$ corresponds to $\phi _ {2}/\phi _ {2}$, and $\phi _ {old}/\phi _ {old}$ to $\phi _ {1}/\phi _ {1}$. Similarly, for the cross-test, $\phi _ {new}/\phi _ {old}$ corresponds to $\phi _ {2}/\phi _ {1}$. In cases of frequent model updates, metrics such as $\phi _ {3}/\phi _ {3}$ and $\phi _ {4}/\phi _ {4}$ are continuously evaluated. As these evaluations accumulate, a variety of model pairs become available for assessing compatibility. Regardless of the notation used these metric evaluations can systematically organized into what is commonly referred to as a compatibility matrix as:
>
> $$
> \left(\begin{array}{cc}
> \phi_{\text{old}}/\phi_{\text{old}} & 0 \\\\
> \phi_{\text{new}}/\phi_{\text{old}} & \phi_{\text{new}}/\phi_{\text{new}}
> \end{array}\right)
> $$
>
> if a further update model $\phi _ {newer}$ is available, it is possible to consider:
>
> $$
> \begin{pmatrix}
> \phi _ {\text{old}}/\phi _ {\text{old}} & 0 & 0 \\\\
> \phi _ {\text{new}}/\phi _ {\text{old}} & \phi _ {\text{new}}/\phi _ {\text{new}} & 0 \\\\
> \phi _ {\text{newer}}/\phi _ {\text{old}} & \phi _ {\text{newer}}/\phi _ {\text{new}} & \phi _ {\text{newer}}/\phi _ {\text{newer}}
> \end{pmatrix}
> $$
>
> In the general case with multiple model update indexed by their sequential time step:
>
> $$
> \begin{pmatrix}
> \phi _ {1}/\phi _ {1} & 0 & 0 & 0 \\\\
> \phi _ {2}/\phi _ {1} & \phi _ {2}/\phi _ {2} & 0 & 0 \\\\
> \phi _ {3}/\phi _ {1} & \phi _ {3}/\phi _ {2} & \phi _ {3}/\phi _ {3} & 0 \\\\
> \phi _ {4}/\phi _ {1} & \phi _ {4}/\phi _ {2} & \phi _ {4}/\phi _ {3} & \phi _ {4}/\phi _ {4}
> \end{pmatrix}
> $$
> All of our metric evaluations are organized into compatibility matrices, as above, and are presented in Figures 7, 8, and 9 of the revised paper for various approaches.
>
> Accordingly, to address the Reviewer's comment accurately, we have replicated the results for all  the approaches in those Figures in the tables below to match the notation used in BT2.
> Specifically, the tables compare our methods—PSP and LSP—with other approaches, namely BCT, CoReS, ETF-CE, ETF-DR, LCE, AdvBCT and BT2, across two datasets: CIFAR100 and TinyImageNet200, in this case in a two-step scenario instead of the multistep one we used in the revised paper.
> All the approaches involves a ResNet18-based model trained on half of the classes available in each dataset, which is then updated to include the entire dataset.
> During the model update of the baseline approach, no compatible methods (e.g. additional losses, mapping, network architecture changes) are used.
> For the baseline, features are extracted from the encoder's output.
> We used BT2 with a feature space of 128, as specified in the original paper, and applied the same settings to BCT and LCE to ensure a fair comparison.
> The results align with those presented in the revised manuscript and reinforce the reproducibility of our analysis across the requested experimental setups. Slight performance variations in BCT and LCE are due to differences in embedding size compared to the manuscript, adjusted to align with the feature size of BT2.

---

> ### Author Response · Authors · 2024-11-29
> **Response to the Official Comment by Reviewer (Zp2R) 3/3**
>
> **TinyImageNet200:**
>
> | **Method**          | **Case**                                          | **CMC (top-1 - top-5)** | **Compatible?** |
> |---------------------|---------------------------------------------------|-------------------------|-----------------|
> | **Baseline**        | $\phi _ {\rm old}/\phi _ {\rm old}$                   | 0.26 - 0.47             |                 |
> |                     | $\phi _ {\rm new}/\phi _ {\rm old}$                   | 0.01 - 0.01             | No              |
> |                     | $\phi _ {\rm new}/\phi _ {\rm new}$                   | 0.38 - 0.61             |                 |
> | **BCT$_{128}$**     | $\phi _ {\rm old}^{\rm BCT}/\phi _ {\rm old}^{\rm BCT}$ | 0.26 - 0.47           |                 |
> |                     | $\phi _ {\rm new}^{\rm BCT}/\phi _ {\rm old}^{\rm BCT}$ | 0.29 - 0.53           | Yes             |
> |                     | $\phi _ {\rm new}^{\rm BCT}/\phi _ {\rm new}^{\rm BCT}$ | 0.41 - 0.62           |                 |
> | **CoReS**           | $\phi _ {\rm old}^{\rm CoRes}/\phi _ {\rm old}^{\rm CoRes}$ | 0.23 - 0.46        |                 |
> |                     | $\phi _ {\rm new}^{\rm CoRes}/\phi _ {\rm old}^{\rm CoRes}$ | 0.22 - 0.44        | No              |
> |                     | $\phi _ {\rm new}^{\rm CoRes}/\phi _ {\rm new}^{\rm CoRes}$ | 0.36 - 0.60        |                 |
> | **ETF-CE**          | $\phi _ {\rm old}^{\rm ETF-CE}/\phi _ {\rm old}^{\rm ETF-CE}$ | 0.24 - 0.47      |                 |
> |                     | $\phi _ {\rm new}^{\rm ETF-CE}/\phi _ {\rm old}^{\rm ETF-CE}$ | 0.24 - 0.47      | No              |
> |                     | $\phi _ {\rm new}^{\rm ETF-CE}/\phi _ {\rm new}^{\rm ETF-CE}$ | 0.39 - 0.61      |                 |
> | **ETF-DR**          | $\phi _ {\rm old}^{\rm ETF-DR}/\phi _ {\rm old}^{\rm ETF-DR}$ | 0.23 - 0.40      |                 |
> |                     | $\phi _ {\rm new}^{\rm ETF-DR}/\phi _ {\rm old}^{\rm ETF-DR}$ | 0.21 - 0.35      | No              |
> |                     | $\phi _ {\rm new}^{\rm ETF-DR}/\phi _ {\rm new}^{\rm ETF-DR}$ | 0.44 - 0.57      |                 |
> | **LCE$_{128}$**     | $\phi _ {\rm old}^{\rm LCE}/\phi _ {\rm old}^{\rm LCE}$ | 0.26 - 0.47          |                 |
> |                     | $\phi _ {\rm new}^{\rm LCE}/\phi _ {\rm old}^{\rm LCE}$ | 0.30 - 0.53          | Yes             |
> |                     | $\phi _ {\rm new}^{\rm LCE}/\phi _ {\rm new}^{\rm LCE}$ | 0.39 - 0.60          |                 |
> | **AdvBCT$_{128}$**  | $\phi _ {\rm old}^{\rm AdvBCT}/\phi _ {\rm old}^{\rm AdvBCT}$ | 0.24 - 0.45      |                 |
> |                     | $\phi _ {\rm new}^{\rm AdvBCT}/\phi _ {\rm old}^{\rm AdvBCT}$ | 0.23 - 0.46      | No              |
> |                     | $\phi _ {\rm new}^{\rm AdvBCT}/\phi _ {\rm new}^{\rm AdvBCT}$ | 0.27 - 0.51      |                 |
> | **BT$^2$$_{128}$**  | $\phi _ {\rm old}^{\rm BT^2}/\phi _ {\rm old}^{\rm BT^2}$ | 0.26 - 0.47        |                 |
> |                     | $\phi _ {\rm new}^{\rm BT^2}/\phi _ {\rm old}^{\rm BT^2}$ | 0.31 - 0.55        | Yes             |
> |                     | $\phi _ {\rm new}^{\rm BT^2}/\phi _ {\rm new}^{\rm BT^2}$ | 0.43 - 0.62        |                 |
> | **PSP (Ours)**      | $\phi _ {\rm old}^{\rm PSP}/\phi _ {\rm old}^{\rm PSP}$ | 0.23 - 0.41         |                 |
> |                     | $\phi _ {\rm new}^{\rm PSP}/\phi _ {\rm old}^{\rm PSP}$ | 0.25 - 0.47         | Yes             |
> |                     | $\phi _ {\rm new}^{\rm PSP}/\phi _ {\rm new}^{\rm PSP}$ | 0.42 - 0.60         |                 |
> | **LSP (Ours)**      | $\phi _ {\rm old}^{\rm LSP}/\phi _ {\rm old}^{\rm LSP}$ | 0.26 - 0.47         |                 |
> |                     | $\phi _ {\rm new}^{\rm LSP}/\phi _ {\rm old}^{\rm LSP}$ | 0.29 - 0.55         | Yes             |
> |                     | $\phi _ {\rm new}^{\rm LSP}/\phi _ {\rm new}^{\rm LSP}$ | 0.44 - 0.62         |                 |

---

> > ### Comment · Reviewer_Zp2R · 2024-12-02
> > **My final**
> >
> > I think it is too much to have to revamp the whole paper on ALL experiments (all architectures such as ViT, all datasets) to the same metric as all the baselines AND compare with the baselines, so the reviewers can get a good sense of the progress. I am not able to read what has been depicted in this rebuttal, try as I might.
> >
> > I felt that this paper has a good idea, training-free it seems, so if all the experiments can be properly situated, it will be a great paper for the next conference.
> >
> > As it stands, I will keep my rating as is at 3.
> >
> > Thanks to the authors for all the information provided.

---

> > > ### Author Response · Authors · 2024-12-03
> > > **Clarifying Experimental Scope**
> > >
> > > We thank the Reviewer for their comments and understand the concerns about the scope of the revisions. To clarify:
> > >
> > > - We respectfully suggest that it is not necessary to revamp the entire paper's metrics to align with all baselines. The adjustments we made respond to specific Reviewer requests and are confined to Appendices H, J, and K, as well as detailed in the rebuttal in the format specifically requested by the Reviewer. In the latter case, it’s important to note that the information was already included in the paper. The adjustments we made were purely for clarification, specifically implemented to meet the Reviewer’s constructive request.
> > >
> > > - In response to the Reviewer concerns about the extensive revisions required, we agree that experiment requests should be feasible within the discussion period. We respectfully suggest the Reviewer that the scope of such requests be carefully considered, as extensive revisions across all experiments and architectures may not be practical.
> > >
> > > - We appreciate the significant effort the Reviewer has invested in understanding our submission, which may have some inherent complexities, and we value their dedication to engaging with our work. More in general, we appreciate the efforts of all Reviewers, particularly noting the increased rating by Reviewer EQDT. We encourage the Reviewer to examine the comments made by Reviewer EQDT and our response, which highlight the improvements made and the potential impact of our work.
> > >
> > > We hope this response comprehensively addresses the concerns raised and provides further clarity. While we respect the Reviewer current stance on the rating, we sincerely hope that the revisions and explanations presented might encourage a reevaluation.

---

### Official Review · Reviewer_9btr · 2024-11-02

**Soundness:** 3
**Presentation:** 4
**Contribution:** 2
**Rating:** 5
**Confidence:** 2

**Summary:**

The paper proposes two methods, PSP and LSP, for improving the backward-compatible training (BCT), a framework for training embedding models that enables deploying new models without re-indexing existing image collections. Their main contribution is to show that any independently trained model can be made compatible with any other by simply using features derived from softmax outputs. They perform an empirical study and utilize Average Compatibility (AC), Average Accuracy (AA), and Average Compatibility Accuracy (ACA) to evaluate backward compatibility. Lastly, there are empirical experiments conclude that in some cases, PSP and LSP outperform other methods.

**Strengths:**

1. The paper proposes two methods, PSP and LSP, to improve backward-compatible training (BCT), potentially allowing the evaluation of the compatibility of the query and gallery model.

2. The study uses diverse comprehensive metrics, including Average Compatibility (AC), Average Accuracy (AA), and Average Compatibility Accuracy (ACA), to evaluate backward compatibility.

3. Experimental results indicate that PSP and LSP can outperform other methods in certain cases, showing promise in improving backward compatibility.

**Weaknesses:**

Motivation is unclear. The author seems to bypass the training process of BCT. However, the BCT method aims to make the new query model gain more accuracy (see [1], Table 3). In this paper, the author uses three compatibility metrics to evaluate compatibility between query and gallery models but does not illustrate how PSP and JSP improve model performance.

---

[1] Shen, Y., Xiong, Y., Xia, W., & Soatto, S. (2020). Towards backward-compatible representation learning. In Proceedings of the IEEE/CVF Conference on Computer Vision and Pattern Recognition (pp. 6368-6377).

**Questions:**

Overall, this paper requires some clarifications on theory and experiments. Given these clarifications, if provided in an author's response, I would consider increasing the score.

For the theory, there are a few steps that need further clarification, especially regarding novelty.

1. One aspect of the PSP method remains unclear. In Eqn.(2),  the author defines the feature representation $\textbf{h}^k = \\| \textbf{P}_{k,k} \sigma(\textbf{z}^k) \\|\_2$. When defining a metric or distance function $d$, the following conditions should be considered:

- The distance between an object and itself is always zero.
- The distance between distinct objects is always positive.
- Distance is symmetric: the distance from x to y is always the same as the distance from y to x.
- Distance satisfies the triangle inequality: if x, y, and z are three objects, then $d(x,z) \le d(x,y) + d(y,z)$

However, the softmax function is not one-to-one, and for any constant $c$, we have $\sigma (\textbf{z}^k) = \sigma (c \textbf{z}^k)$. This implies that the distance between two different raw representations, $\textbf{z}$ and $c \textbf{z}$ would reduce to zero, thus violating the first rule.

2. The contribution of the LSP method appears potentially controversial. The model backbone output in response to an image is usually referred to as its ``embedding". However, we double-checked the code (**eval\_logits.py** and **run\_softmax.sh**) provided in the supplemental materials and confirmed that the authors use the final output (logits) of pretrained models as the feature representation. In this scenario, two pre-trained models on the same dataset would be naturally compatible. For well-trained models, their prediction probabilities for the same image should align naturally. Thus, discussing this phenomenon may add limited value.


For the experiments, the following should be addressed.

1. The author should explore the applications or advantages of PSP and LSP. For example:
- Model A is trained on both the training and test sets, while Model B is fairly trained on the training set only. PSP or LSP could then be used to perform hypothesis testing on whether ``Model A and Model B are trained on the same image collection''.
- It would have been better also to show the classification accuracy gain of the new query model.

---

> ### Author Response · Authors · 2024-11-21
> **Response To the Highlighted Strengths (9btr)**
>
> We thank the Reviewer for their thoughtful feedback and for recognizing the strengths of our work.

---

> ### Author Response · Authors · 2024-11-21
> **Response To the Weaknesses (9btr)**
>
> Please refer to the Response To the Questions (9btr) for the response.

---

> ### Author Response · Authors · 2024-11-21
> **Response To the Questions of Reviewer (9btr)**
>
> [**Clarification about Norm vs Normalization in Eq. 2.**]
>
> We thank the Reviewer for the clarification regarding Equation (2) and the definition of the feature representation $\mathbf{h}^k$. We inadvertently made a typo by missing the numerator, which may have misled the Reviewer and made the equation incomplete. We apologize for this oversight. The correct formula is:
>
> $ \mathbf{h}^k = \frac{ \mathbf{P} _ {k, k} \  \sigma( \mathbf{z}^k )}{ \| \| \mathbf{P} _ {k, k} \  \sigma(\mathbf{z}^k) \| \|_2}.$
>
> Therefore, to clarify, $ \mathbf{h}^t $ is a normalized feature vector, not the result of a norm we inaccurately described as yielding a scalar for distance measurement.
>
>  ---
>
> [**Code Double Check and Contribution.**]
>
> We thank the Reviewer for the careful review and for taking the time to check the code provided in our supplemental materials. We appreciate the Reviewer's efforts in verifying that logits and the probability outputs from the pretrained models are indeed used as feature representations, as described. We again apologize for our oversight which may have misled the Reviewer.
>
> We completely agree with the Reviewer that for well-trained models, the prediction probabilities for the same image should naturally align. Our contribution is to provide a formal proof of this intuition, which not only confirms the phenomenon but also strengthens its theoretical foundation.
>
> ---
>
> [**Possible Applications and Advantages.**]
>
> We thank the Reviewer for their insightful comments, which indeed seem to align with the framework of compatibility learning, originally defined in [1] to evaluate the compatibility of models trained under different conditions.
>
> This raised issue may resemble the ''Neural Lineage'' problem recently presented at CVPR2024, which involves identifying the parent model from which a well-behaved neural network was fine-tuned.
> Specifically, neural lineage detection involves predicting from which parent model a child model has been fine-tuned, given a set of potential parent models.
> If we understand correctly what the Reviewer pointed out, it may share similarities with the ''Neural Lineage''. However, instead of detecting if a model was obtained by fine-tuning from a parent model, in the case of compatible representations, one should determine if the base model was learned using the same training dataset similar to another model.
> The point raised by the Reviewer is very interesting and merits future research into its implications.
>
> [1] Shen, Y., Xiong, Y., Xia, W., and Soatto, S. (2020). Towards backward-compatible representation learning. In Proceedings of the IEEE/CVF Conference on Computer Vision and Pattern Recognition (pp. 6368-6377).
>
> ---
>
> [**Accuracy Gain of the New Model.**]
>
> We appreciate this observation. Here, the task of classification is used as a ''surrogate task'' to learn the internal feature representation, which is then employed to represent images for retrieval/search tasks. Since the number of classes in the surrogate classification task varies at each step, the gain in classification accuracy does not directly provide a meaningful measure of how well images are represented. For instance, the first surrogate classification task may involve 5 classes and the subsequent one 20 classes. The first might show higher classification accuracy due to its simplicity, while the second, a more challenging task, might show lower accuracy. However, the features learned from the second task, involving 20 classes, are more expressive for representing images despite the reduced classification accuracy, as they incorporate a greater diversity of data into the representation.
>
> We appreciate the Reviewer's intuition regarding the gains derived from the classifier's surrogate task. If we understand the question correctly, the main diagonals of the compatibility matrices support this intuition. These diagonals quantify the performance gains achieved when a learned model is used for retrieval, as both the query and gallery are extracted with the same model. This directly demonstrates the impact of the surrogate classification task on the learned representation.
>
> To clarify, we have included the compatibility matrices for PSP, LSP, and all compared methods in Appendix H of the revised manuscript, specifically for the 5-step scenario on the CIFAR100 dataset under the extended classes setting.

---

> ### Comment · Reviewer_9btr · 2024-11-25
>
> Thank you for providing further clarification. Based on your response, I summarize my understanding of your work:
>
> >**[Clarification about Norm vs Normalization in Eq. 2.]** We thank the...
>
> The authors may have misunderstood my first question. If we aim to compute the similarity between vectors $z_1$ and $z_2$, we cannot define a many-to-one mapping:
>
> $$\psi: z \mapsto h,$$
>
> and then compare $h_1=\psi(z_1)$ and $h_2=\psi(z_2)$ to measure the similarity between $z_1$ and $z_2$.
>
> For example:
>
> $$h=\psi(z)=\frac{z}{\\|z\\|}, z_1 = (10000, 0), z_2 = (1, 0)$$
>
> It is evident that the features $z_1$ and $z_2$ are distinct. However, after applying the many-to-one mapping, $h_1=\psi(z_1)$and$h_2=\psi(z_2)$ become identical.
>
> >**[Accuracy Gain of the New Model.]** We appreciate this...
>
> So, the primary application appears to be task-specific, specifically for surrogate classification. Am I correct?

---

> > ### Author Response · Authors · 2024-11-27
> > **Response to Post Rebuttal discussion (9btr)**
> >
> > We thank the Reviewer for the feedback and apologize for any misunderstanding of their initial question.
> >
> > ---
> >
> > **[Clarification about Norm vs Normalization in Eq. 2.]**
> >
> > We appreciate the opportunity to clarify that the many-to-one mapping is indeed a deliberate advantage of employing cosine distance in the hypersphere. After normalization, all distinct representations can be compared based on the subtended angles according to the cosine distance. The use of this distance is formally justified by the fact that the representations especially for search and retrieval are typically directly learned in the hyperspherical domain. This is typically achieved by setting the classifier bias to zero, which eliminates the need for using Euclidean distance in Euclidean space. It has been demonstrated in SphereFace [1] that angles correlate with semantics and that biases are typically close to zero. The paper has been influential in demonstrating that learning these biases do not improve the learned representation. Moreover, it demonstrates that search and retrieval using representations learned with a classifier that set these biases to zero validate the effectiveness of cosine distance, as the representations are directly learned in the hyper-spherical manifold.
> >
> > ---
> >
> > **[Accuracy Gain of the New Model.]**
> >
> > Regarding the use of classification as a surrogate task, we clarify that this is not a limitation and does not restrict the audience for our proposed methodology.
> > Fine-tuning a classifier on a specific distribution, starting from a previously learned representation, is a well-established and highly effective approach for retrieval task, particularly in open-set scenarios [2].
> > Models trained in a self-supervised manner learn general-purpose representations, which often require adaptation to specific tasks such as retrieval or classification. We acknowledge the Reviewer's point that classification relies on supervised learning and, therefore, necessitates labeled data, which may not always be readily available or could be costly to obtain.
> > Although our proposed method utilizes labels, it is both a theoretical and practical approach that can be adapted for use in low-label regimes by utilizing self-supervised learning to pre-train the model and establish a robust internal representation.
> > For instance, we can leverage fine-tuning to build an updated model by starting with an independently self-supervised trained model and then fine-tuning it with supervision, thereby potentially requiring fewer data. This flexibility ensures that our method can be applied across a wide range of learning scenarios.
> >
> > As an example, in the experiments with ViT architectures trained on ImageNet1k added during this rebuttal (Appendix J), the ViT-L-16 is fine-tuned starting from a self-supervised pretrained model [3], which provides an internal representation learned through self-supervision, rather than learning the representation from scratch via a (surrogate) classification task, as in the case of ViT-B-32 and ViT-B-16.
> > In this case, our method leverages fine-tuning from an updated model, whose internal representation was learned through self-supervised learning.
> >
> > [1] Liu, W., Wen, Y., Yu, Z., Li, M., Raj, B., and Song, L. "Sphereface: Deep hypersphere embedding for face recognition". (CVPR2017).
> >
> > [2] Vaze, S., Han, K., Vedaldi, A., and Zisserman, A. Open-Set Recognition: "A Good Closed-Set Classifier is All You Need". (ICLR2021).
> >
> > [3] Singh, M., Gustafson, L., Adcock, A., de Freitas Reis, V., Gedik, B., Kosaraju, R. P., ... and Van Der Maaten, L. "Revisiting weakly supervised pre-training of visual perception models". (CVPR2022).

---

> ### Comment · Reviewer_9btr · 2024-12-02
>
> Thank you for the detailed clarification. I am still confused about the evaluation metric used in this paper. If the method demonstrates accuracy improvement and focuses on the classification task, why not use Top-1 Accuracy? Conversely, if the method focuses on a retrieval task, why not use metrics such as Precision, Recall, F1-Score, or Mean Average Precision? I am not entirely familiar with this domain (AC, AA, ACA) and therefore defer to the ACs and other reviewers who may have more expertise in this area.

---

> ### Author Response · Authors · 2024-12-03
> **Clarifying Evaluation**
>
> We thank the Reviewer for the follow-up questions and for expressing their thoughts on the evaluation metrics. To clarify:
>
> 1. Our study addresses retrieval task in model compatibility. No classification task.
> 2. Model compatibility requires evaluating retrieval performance (we used Recall@1) across different model versions: the gallery is expressed with model $\phi_1$ while queries with model $\phi_2$.
> 3. As the number of sequential model updates increases (i.e., $\phi_1$, $\phi_2$, $\phi_3$, ...), so does the number of paired interactions between query and gallery expressed with these models.
> 4. The AC, AA, and ACA metrics summarize the performance of paired model interactions into three numbers by averaging all interactions, as detailed in Section 4.1.
>
> We acknowledge the inherent complexity of these metrics and appreciate the Reviewer's trust in the expertise of the Area Chair and other Reviewers.

---

### Official Review · Reviewer_dqbF · 2024-11-04

**Soundness:** 3
**Presentation:** 2
**Contribution:** 3
**Rating:** 5
**Confidence:** 3

**Summary:**

This paper studies the backward compatible ability between the representations of the base model and the updated ones. The study shows that independently trained neural networks can be made compatible simply using feature representations derived from softmax outputs and logits. Experiments on Cifar100, TnyImageNet200, ImageNet1k and Google Landmark v2 are conducted to demonstrate the effectiveness of the proposed method.

**Strengths:**

-	The paper includes comprehensive theoretical analysis.
-	The performance decay is clear on previous methods when the number of steps increases. The proposed method achieves interesting results, even with large number of training steps.

**Weaknesses:**

-	The theoretical analysis appears complex and challenging to follow. Could the authors clarify how to define the projection matrix $P_{k,k}$.
-	It would be better to add some pseudo codes on LSP and PSP, which could help readers understand how to implement the proposed methods.
-	In Table 1, it is interesting to see the old method BCT works better than new ones on most of cases. Why newer methods, such as CoRes and ETF-CE, show lower performance than BCT?
-	BCT is mainly evaluated on face recognition datasets. Does the proposed methods LSP and PSP also work on face recognition datasets?

**Questions:**

Please see above.

---

> ### Author Response · Authors · 2024-11-21
> **Response To the Highlighted Strenghts (dqbF)**
>
> We thank the Reviewer for their positive comments regarding the comprehensive theoretical analysis in our manuscript and for recognizing our method’s robust performance across a large number of model update steps. We appreciate the insightful feedback.
> We also thank the Reviewer for the feedback regarding the complexity of the theoretical analysis in our manuscript.

---

> ### Author Response · Authors · 2024-11-21
> **Response to the Weaknesses (dqbF) 1/2**
>
> [**Clarification of the Projection Matrix Definition is needed due to Complex Analysis.**]
>
> We appreciate the Reviewer's comment for clarification on the definition of the projection matrix $\mathbf{P} _ {k,k}$.
> To provide a more concise explanation, the projection matrix $\mathbf{P} _ {k,k}$ can be succinctly expressed using the concept of the centering matrix \cite{marden1996analyzing}, defined as:
> $$\mathbf{P} _ {k,k}  = \mathbf{I}  _ {C^k} - \frac{1}{C^k} \mathbf{J} _ {C^k}$$
> where $ \mathbf{I} _ {C^k} $ is the identity matrix, $ C^k $ represents the number of classes, and $ \mathbf{J}_{C^k} $ is a matrix entirely composed of ones. The centering matrix is typically used to remove the mean, thus centering the data.
> In our case, it is specifically applied to the vertices of the Probability Simplex $\mathbf{e}_i$ stacked into the identity matrix $ \mathbf{I} _ {C^k} $, while the matrix $ \frac{1}{C^k} \mathbf{J} _ {C^k} $ subtracts the mean to each vertex $\mathbf{e}_i$.
> We have included this clarification in the revised manuscript. We thank the Reviewer for their suggestions, which have improved the readability and understanding of our work and provided us with a different perspective.
>
>
> ---
>
> [**Pseudo-code of Proposed Methods.**]
>
> We thank the Reviewer for the constructive suggestion. We agree that this addition will improve the paper. We have appended pseudo code in the revised manuscript in Appendix F.
>
> ---
>
> [**Newer Methods Performance.**]
>
> Newer Algorithms such as CoReS and ETF-CE may perform worse when the model is updated for few steps because they are designed to enforce a fixed predetermined representation space. They are designed to pre-allocate a large number of future classes to accommodate any further class data expansion. Although ETF-CE is not specifically designed for compatibility, it performs learning through a simplex fixed classifier as in CoReS. In principle, this approach is limited by the number of class prototypes in the classifier, which must be fewer than the encoder's output size. CoReS addresses this issue by adding a linear layer before the classifier to extend this limit. Conversely, BCT operates more adaptively by enforcing classifiers that have been learned in the past and then kept fixed.
> BCT is based on what the authors term ''transitive compatibility'', which suggests compatibility is a pairwise relationship between models and not a global one as formulated in CoReS and ETF-CE. This pairwise model relationship used in BCT can extend to multiple model updates, but only under the assumption that the transitive property holds. As an example BCT allows the final model, $\phi _ 3$, to be transitively compatible with $\phi _ 1$, even though $\phi _ 1$ is not directly involved in the training of $\phi _ 3$. Therefore, there is a trade-off between these two strategies, depending on whether long-term or short-term compatibility is required.
> In light of this basic working principle, the good performance of BCT may be attributed to the fact that neither ETF-CE nor CoReS leverage the previous model.
> As a final note, in the experiment we used the more computationally intensive variant of BCT that uses ''synthesized classifier weights''. In this variant, the classifier weights for the previous step are partially computed as the mean of the features extracted by the previous model from the images of the novel step. This process leverages the old model to extract features from the new training set, maximizing its utility. In contrast, the standard BCT implementation leads to a substantial performance drop, as demonstrated in the table below. Specifically, the performance metrics AC, AA, and ACA of the standard BCT across models trained for 2, 5, 20, and 50 steps on CIFAR100 are presented, and evaluated using Recall@1 on the CIFAR100 test set. A comparison of these results with the BCT performance in Tab. 1a of the submitted manuscript demonstrates the superior performance of the BCT variant. However, the naive implementation involves significantly less computational effort, as it relies only on the old classifier rather than the entire old model.
>
> | Method   | 2 steps AC | 2 steps AA | 2 steps ACA | 5 steps AC | 5 steps AA | 5 steps ACA | 20 steps AC | 20 steps AA | 20 steps ACA | 50 steps AC | 50 steps AA | 50 steps ACA |
> |----------|------------|------------|-------------|------------|------------|-------------|----------|------------|------------|-------------|------------|------------|
> | BCT      | 0          | 36.61      | 0           | 0.10       | 25.71      | 3.21       | 0.1        | 18.98       | 3.90       | 0.03 | 15.33 | 1.25 |

---

> ### Author Response · Authors · 2024-11-21
> **Response to the Weaknesses (dqbF) 2/2**
>
> [**Face Recognition Evaluation.**]
>
> We appreciate the reviewer’s comment regarding the applicability of our proposed methods, LSP and PSP, to face recognition datasets. In response to this, we have conducted evaluations of both methods using the CASIA-WebFace dataset [1] and the IJB-C 1:1 verification protocol [2].
> Results confirm that our methods consistently achieve the highest performance in scenarios involving multiple model updates, and they perform comparably to BCT when the model is updated once.
> Importantly, while there is a slight decrease in average accuracy ($AA$), we observe a significant gain in compatibility with BCT as the number of update steps increases.
>
> [1] Dong Yi, Zhen Lei, Shengcai Liao, and Stan Z Li. Learning face representation from scratch. arXiv preprint arXiv:1411.7923, 2014.
>
> [2] Brianna Maze, Jocelyn Adams, James A Duncan, Nathan Kalka, Tim Miller, Charles Otto, Anil K Jain, W Tyler Niggel, Janet Anderson, Jordan Cheney, et al. Iarpa janus benchmark-c: Face dataset and protocol. In 2018 international conference on biometrics (ICB), pp. 158–165. IEEE, 2018.
>
> | Method   | 2 steps AC | 2 steps AA | 2 steps ACA | 5 steps AC | 5 steps AA | 5 steps ACA |
> |----------|------------|------------|-------------|------------|------------|-------------|
> | Baseline | 0          | 50.20      | 0           | 0          | 36.72      | 0           |
> | BCT      | 1          | 72.32      | 68.88       | 0.40       | 70.91      | 28.69       |
> | PSP      | 1          | 67.32      | 65.21       | 1          | 65.24      | 59.78       |
> | LSP      | 1          | 71.61      | 68.26       | 0.70       | 72.34      | 58.47       |

---

### Author Response · Authors · 2024-11-21
**Acknowledgements and Revisions**

We thank the Reviewers for the effort that they invested into our manuscript and the Area Chair for reading the reviews and our rebuttals. We have carefully considered each point raised and have made corresponding revisions to the manuscript.

---

> ### Author Response · Authors · 2024-11-22
> **Response Summary**
>
> In our responses, we addressed key concerns and provided clarifications to highlight the impact and relevance of our work. We pointed out the balance between achieving backward compatibility and maintaining strong model performance, demonstrating the practical benefits in scenarios like infrequent model updates and discussed the advantages of the frequent ones. To further strengthen our methodology, we explored top-k dimensionality reduction, showing its utility in learning compatible representation, evaluated on face recognition tasks and integrated PSP/LSP features into BCT. We discussed compatibility in self-supervised learning and the theoretical limit as the number of updates approaches infinity, reinforcing the soundness of our approach. To ensure clarity, we improved the presentation of the paper, incorporating pseudo-code of the proposed methods and additional results with Vision Transformer architectures. We also elaborated on the accuracy gains achieved by our new model and addressed theorem assumptions and their implications.

---

> > ### Author Response · Authors · 2024-12-01
> > **Post Rebuttal Response Summary**
> >
> > In the post-rebuttal discussion, **Reviewer EQDT** **positively acknowledged** the proposed approach to model compatibility, noting its **theoretical support** and **lack of training** requirement. However, **concerns** were raised about the method's **practicality** and theoretical basis due to **high feature vector dimensionality** and its performance in frequent update scenarios. The authors responded by introducing **top-k dimensionality reduction**, **supported** by **theoretical insights** from the Lapin et al. paper, to effectively manage feature size. Empirical evidence demonstrates the method's **robustness** across various update steps and with **large numbers of classes**, thereby improving its practical application. The Reviewer also commented on the possibility of **partial backfilling**, which the authors demonstrated as a possible efficient update management technique in their method. Finally, the Reviewer **positively influenced** the **contributions** of the method outlined at the end of the introduction section.
> >
> > **Reviewer Zp2R** also **acknowledged positively** the proposed work and requested that the authors **compare** PSP/LSP **directly** against established methods like BCT and BT2 using **traditional** before/after **metrics** in a standard table format. The authors acknowledged the confusion and clarified that their metrics, although presented differently, **corresponded** to the **traditional formats**. They provided their presentation to **align** with the **conventional table** format used in previous studies **to ensure clarity** and comparability. The interaction with the **Reviewer** **helped** the authors **better understand** how to introduce the **complex evaluation** of compatible learning. They now more effectively **justify** the use of metrics AC, AA, and ACA for the **trade-off** between accuracy and compatibility in frequent update scenarios. They also noted the use of **self-supervised** learning in **ViT** architectures, which the proposed method can implicitly leverage.
> >
> > The authors expressed gratitude to **Reviewer dqbF** for **initial feedback** that **improved** the PSP presentation and for recommending face recognition tests. They also thanked **Reviewer 9btr**, whose **initial feedback** directly led to significant **improvements** in the **revised manuscript**.
> > The **authors remain eager** to address any **additional questions** the Reviewers may have, which could assist in **finalizing their ratings** of the paper.

---

> ### Author Response · Authors · 2024-12-03
> **Final Summary of Changes**
>
> These changes represent a considerable additional effort to address Reviewer concerns, clarify ambiguities, and include numerous model architectures. This effort not only demonstrates the method’s applicability but also confirms its theoretical support for handling more complex datasets and architectures.
>
> Changes include:
> - **Appendix F**: Pseudo-code for PSP/LSP with softmax or logits. Thanks to Reviewer ``dqbF``.
> - **Appendix H**: Compatibility Matrices for CIFAR100. Thanks to Reviewer ``Zp2R``.
> - **Appendix I**: Compatibility Matrices for ImageNet1k pre-trained models: added Places365. Thanks to Reviewer ``Zp2R``.
> - **Appendix J**: ViT Architectures, similar to Appendix I, with external pre-training. Thanks to Reviewer ``Zp2R``.
> - **Appendix K**: Top-K sparsification. Thanks to Reviewer ``EQDT``.
> - **Section 1**: Improved contributions as suggested by Reviewer ``EQDT``.
> - **Section 3**: Listed assumptions and theorem results. Thanks to Reviewer ``EQDT``.
> - **Section 3**: Improved presentation using centering matrix. Thanks to Reviewers ``dqbF``, ``Zp2R``.
> - **Section 4**: Revised the limitations. Thanks to Reviewer ``EQDT``.
> - **Section 4**: Improved explanation about metrics. Thanks to Reviewer ``Zp2R``.
>
> We believe these updates have significantly improved the quality of our paper. We sincerely thank all Reviewers for their constructive feedback and active participation throughout this discussion phase.
>
> In addition to these major changes, we have addressed all minor concerns raised by the Reviewers and actively engaged in the discussion to address their questions.
> A summary of the responses, pre and post-rebuttal, can be found in the [Response Summary](https://openreview.net/forum?id=pPmQvd1NUp&noteId=60Z3ZnoAVQ) and [Post Rebuttal Response Summary](https://openreview.net/forum?id=pPmQvd1NUp&noteId=r7eJaOZYqU), respectively.
>
> All changes are highlighted in blue in the revised PDF, which is available through the usual link on OpenReview.

---

### Meta-Review · Area_Chair_3m1u · 2024-12-20

**Metareview:**

The reviewers acknowledge the benefits of this work, including its interesting ideas and method analysis, and they believe it could serve as a valuable study in the field. However, as commonly mentioned in the initial reviews, understanding the paper was difficult due to its unclear presentation, confused writing, and ambiguous evaluation metrics. The reviewers raised various questions about motivation, experiments, and metrics, which the authors diligently addressed in their rebuttal.

While the authors made some effort to address the reviewers' concerns, the current version of the paper still falls short of publication standards, as pointed out by Reviewer Zp2R. Additionally, the AC believes that making too many changes and additions during the rebuttal period is not ideal. Therefore, the AC recommends submitting this work with current improvements to a future conference/journal rather than this ICLR.

**Additional Comments On Reviewer Discussion:**

Reviewer dqbF, Reviewer 9btr, and Reviewer Zp2R raised concerns about the presentation of the paper and writing quality. Reviewer Zp2R specifically noted that the experiments in the submitted version of the paper failed to properly show the proposed method's strengths, and also mentioned the necessity of more comprehensive experiments. The authors diligently provided point-to-point responses to the reviewers' questions. Reviewer dqbF, Reviewer 9btr, and Reviewer Zp2R still believe the paper needs significant improvement to be published and maintain their rejection recommendations.

---

### Decision · Program_Chairs · 2025-01-22

Reject